# Biomimetic α-selective ribosylation enables two-step modular synthesis of biologically important ADP-ribosylated peptides

Anlian Zhu[1,4], Xin Li [2,4], Lili Bai[1,4], Gongming Zhu[1], Yuanyang Guo [1], Jianwei Lin[2], Yiwen Cui[2], Gaofei Tian[2], Lihe Zhang[3], Jianji Wang[1], Xiang David Li [2✉] & Lingjun Li[1✉]

The α-type ADP-ribosylated peptides represent a class of important molecular tools in the field of protein ADP-ribosylation, however, they are difficult to access because of their inherent complicated structures and the lack of effective synthetic tools. In this paper, we present a biomimetic α-selective ribosylation reaction to synthesize a key intermediate, α-ADP-ribosyl azide, directly from native β-nicotinamide adenine dinucleotide in a clean ionic liquid system. This reaction in tandem with click chemistry then offers a two-step modular synthesis of α-ADP-ribosylated peptides. These syntheses can be performed open air in eppendorf tubes, without the need for specialized instruments or training. Importantly, we demonstrate that the synthesized α-ADP-ribosylated peptides show high binding affinity and desirable stability for enriching protein partners, and reactivity in post-stage poly ADP-ribosylations. Owing to their simple chemistry and multidimensional bio-applications, the presented methods may provide a powerful platform to produce general molecular tools for the study of protein ADP-ribosylation.

[1] Henan Key Laboratory of Organic Functional Molecule and Drug Innovation, School of Chemistry and Chemical Engineering, Key Laboratory of Green Chemical Media and Reactions, Ministry of Education, Henan Normal University, 453007 Xinxiang, Henan, China. [2] Department of Chemistry, The University of Hong Kong, Pokfulam Road, Hong Kong, China. [3] State Key Laboratory of Natural and Biomimetic Drugs, School of Pharmaceutical Science, Peking University, 100191 Beijing, China. [4] These authors contributed equally: Anlian Zhu, Xin Li, Lili Bai. ✉email: xiangli@hku.hk; lingjunlee@htu.edu.cn

ADP-ribosylation, a multifaceted posttranslational modification, in which ADP-ribose (ADPr) units are transferred from β-nicotinamide adenine dinucleotide (β-NAD⁺) onto site-specific amino acids of proteins, is involved in a wide range of physiologic processes in health and disease (Fig. 1a, b)[1,2]. In most cases, ADP-ribosylation is a highly dynamic and reversible process, and ADPr units can be easily removed from the generated ADP-ribosylated proteins in an enzyme-dependent or spontaneous manner[3]. Although more than 900 ADP-ribosylated modification sites have been identified[4], understanding of the mechanisms of ADP-ribosylation processes is still lagging[1–5]. The limitations of the existing toolbox and the lack of effective molecular tools greatly hamper biological research in this field. Therefore, it is of great need to artificially synthesize ADP-ribosylated peptides that are stable in cellular environments and can mimic their physiological functions, such as ligands in binding protein models[6], as probes for enriching and capturing endogenous partners[7], and as a key substrates for preparing antibodies of site-specific ADP-ribosylated proteins, which has largely been hindered by the lack of effective antigens[8].

However, the synthesis of well-defined ADP-ribosylated peptides is rather challenging because of their inherent complex structures. An α-type glycosidic bond is formed at the 1″ position of the free nicotinamide ribose via an SN1-like enzymatic reaction mechanism (Fig. 1a)[9], which is difficult to realize by chemical synthesis. This synthesis needs to stereoselectively construct glycosidic bonds in the presence of free peptides and ADPr motifs contain highly vulnerable pyrophosphates, naked ribosyl moieties, and adenine ring with multiple N-nucleophilic and O-nucleophilic sites. Indeed, Vorbruggen-type ribosylations are the only reactions that are currently used in such syntheses (Fig. 1c). In a typical synthetic route of ADP-ribosylated peptides[10–12], the construction of α-type nucleosides via Vorbruggen-type reactions requires harsh acidic catalysis, delicate directing groups, and multiple protective groups, which inevitably results in long and multistep synthetic routes involving highly moisture-sensitive and low-yield reactions like pyrophosphorylations. Such syntheses can only be performed by specialists in highly specialized laboratories.

Alternatively, a bioconjugation method was developed using an oxime-ligation to form glucosidic bonds (Fig. 1d)[13], but the major products are open-ring structures resulting in α/β-type mixtures of ADPr derivatives that do not have the features of α-type glycosidic bonds, which are crucial for the physiological functions of ADP-ribosylated peptides (Fig. 1b).

In this paper, we demonstrate a biomimetic α-selective ribosylation reaction that enables the two-step modular synthesis of biologically important α-ADP-ribosylated peptides (Fig. 1e). The route contains only two reactions in clean aqueous ionic liquid catalytic systems, which can easily be performed open air in eppendorf (EP) tubes without specialized equipment or training. We were able to achieve large-scale preparation of biomimetic ribosylation (at 0.5-g scales) in high yields (86% for biomimetic ribosylation; >82% for total yields of target peptides). More importantly, the α-ADP-ribosylated peptides prepared by this method exhibited high binding affinity and desirable stability in enriching endogenous protein partners, and reactivity in enzymatic post-stage poly ADP-ribosylations. Collectively, our approach advances the synthesis of ADP-ribosylated peptides owing to its simple chemistry and multipurpose biological applications, which can facilitate the design and preparation of better general molecular tools for the study of protein ADP-ribosylation.

## Results

**Biomimetic α-selective ribosylation reactions.** In nature, different amino acids show synergistic effects to promote protein α-selective ribosylations using β-NAD⁺[14]. For example, in the catalytic domains of ribosyltransferases, acidic Glu or Asn residues offer hydrogen bond interactions for the activation of hydroxyl groups and nicotinamides, while basic Arg residues provide electrostatic interactions with pyrophosphate motifs to orientate preferential conformations (Fig. 2a). However, it is hard to mimic the enzymatic catalytic processes using artificial reaction systems. The kinetics of β-NAD⁺ hydrolysis reactions have been found to be affected by pH, temperature and nucleophiles;[15] nevertheless,

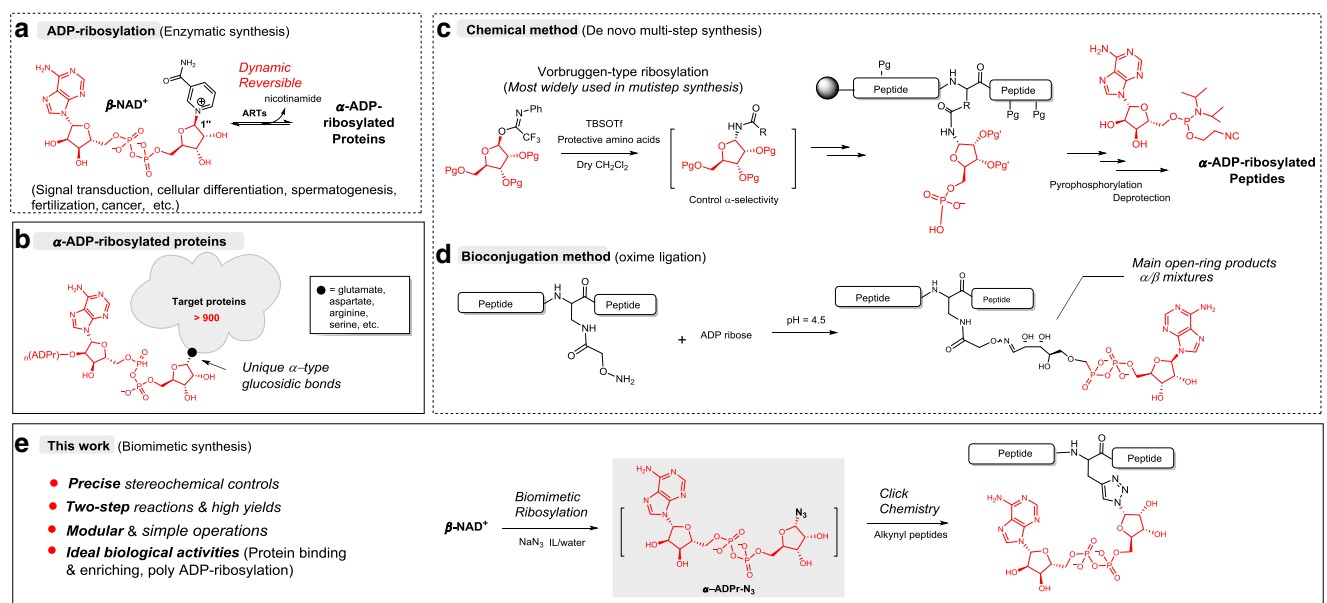

**Fig. 1 ADP-ribosylations and strategies for synthesizing α-type ADP-ribosylated peptides. a** The enzymatic synthesis route for ADP-ribosylations. **b** The general chemical structure of α-ADP-ribosylated proteins. **c** The chemical synthesis route for ADP-ribosylated peptides. **d** The bioconjugation method for syntheses of ADP-ribosylated peptides. **e** The biomimetic route for syntheses of ADP-ribosylated peptides that was shown in this work. ADP adenosine diphosphate, ARTs ADP-ribosyltransferases, Pg protecting group, ADPr ADP-ribose, IL ionic liquid.

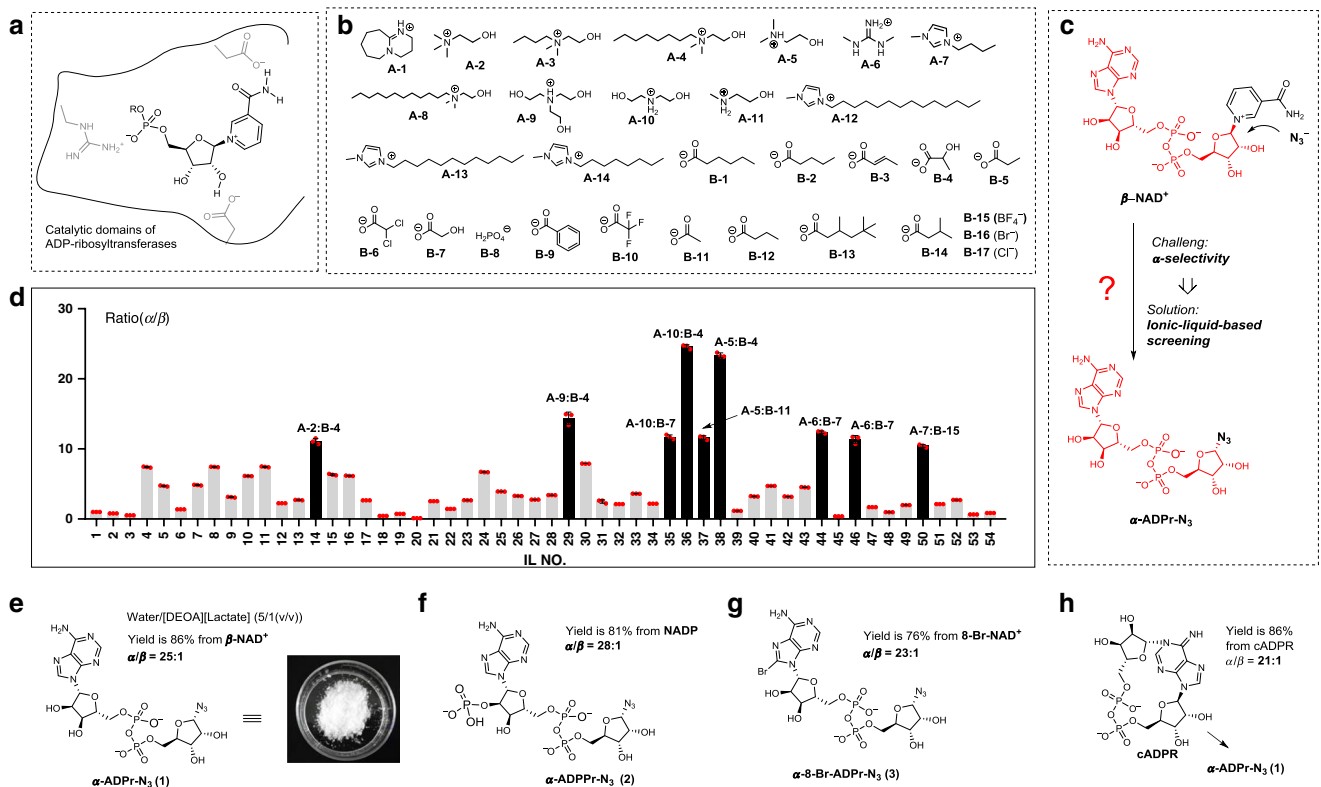

**Fig. 2 The biomimetic α-selective ribosylation reactions are developed by using ionic-liquid-based screening. a** Synergistic effects in enzymatic ADP-ribosylation reactions. **b** Cations and anions of ionic liquids for screening. **c** Biomimetic transformations of β-NAD+ to α-type ADP-ribosyl azide. **d** Results of screening on 54 ionic liquids; black bars show the ionic liquids with high α-selectivity (the ratios of α/β-isomer more than 10:1). Experiments were performed in triplicates. Data (available at https://doi.org/10.6084/m9.figshare.12375032) are reported as mean ± s.d. ($n = 3$), and black dots indicate value of each sample. **e** Best reaction system for the synthesis of α-ADPr-N3 (**1**). **f** Biomimetic synthesis of α-ADPPr-N3 from β-NADP. **g** Biomimetic synthesis of 8-Br-α-ADPr-N3 from 8-Br-β-NAD+. **h** Biomimetic synthesis of α-ADPr-N3 from cyclic ADP-ribose (cADPR).

no optimized reaction conditions can be used for preparative purposes, especially for generating α-type ADPr products. Indeed, stereocontrol is lacking in all the chemical reactions of β-NAD+[15–19]. Ionic liquids (IL) are a class of liquid materials composed of cations and anions that can be applied in various chemical processes. Proper pairing of cations and anions in ionic liquids can produce synergistic catalytic effects on chemical transformations that would hardly proceed under mild conditions using traditional reaction procedures[20–24]. In addition, the easy combination of cations and anions leads to a wide variety of ionic liquids up to $10^{18}$, which offer opportunities to apply screening-based methods on ionic liquid libraries for the discovery of catalytic procedures, when no rich information about the reaction mechanisms is available[24]. Notably, some ionic liquids bearing rich noncovalent interactions, particularly hydrogen-bonding donors/acceptors that have been demonstrated to be able to regulate and control the spatial conformation of biomolecules[25–28] or stabilize active reaction intermediates[29], have been used in a series of potential biomimetic catalytic systems[30]. All these merits inspired us to investigate, whether anion and cation combinations in ionic liquids could offer suitable reaction systems for promoting α-selective ribosylation of β-NAD+.

Figure 2b shows a library of ionic liquids with various cations and anions that have been used in the discovery of biomimetic reactions for transferring β-NAD+ into α-ADPr-N3 (**1**) (Fig. 2c). After optimizing the water content of the reaction (Supplementary Table 1 in Supplementary Information), screening was performed in aqueous solutions of the ionic liquids (water:IL, 5:1 mol/mol). The α-selectivity of 54 ionic liquids (Fig. 2b and Supplementary Table 2 in Supplementary Information) was

sorted by HPLC analysis (Fig. 2d and Supplementary Table 3, Supplementary Figs. 2–7 in Supplementary Information). Dramatically, in the [DEOA][Lactate] (A-10: B-4 in Fig. 2d) IL aqueous solution, the ratio of α/β-isomer was desirably up to 25:1 (with the selectivity of α-isomer over 96%). For preparative purposes, the scale-up of the α-ADPr-N3 experiment was conducted in the [DEOA][Lactate] IL/water reaction system (Supplementary Table 4 in Supplementary Information), which resulted in an 87% yield at the 0.5-g scale (Fig. 2e). Different nucleotide substrates were then tested in the reaction (Fig. 2f–h). Using the [DEOA][Lactate] IL/water reaction system, we obtained a 77% yield for α-ADPPr-N3 (**2**) from β-NADP after 2 h (Fig. 2f). A few ADP-ribosyltransferases can recognize NADP as a substrate[31], our biomimetic method thus provides a very useful approach for the synthesis of α-ADPr phosphate derivative. The 8-Br-NAD+ is a class of substrate that can be transferred to diversely modified ADP-ribosylated molecules with attractive bioactivities[32]. We also found it to be an effective reactant for this reaction giving an 81% yield for the target 8-Br-α-ADPr-N3 product (Fig. 2g). This provides the possibility of using biomimetic ribosylation to synthesize 8-substituted α-ADPr derivatives. Besides nicotinamide, we investigated other different leaving groups such as cyclic ADP-ribose (cADPR)[33] (Fig. 1h). Using a potential open-ring reaction between the north ribosyl moiety and $N - 1$ of the purine base, we obtained a 65% yield for α-ADPr-N3 from cADPR. Conventionally, harsh anhydrous treatments and inert gas atmospheres are required for chemical ribosylation reactions, because of the presence of highly active directing groups and intermediates[11,12]. It is worth-mentioning that no anhydrous treatments are used in our biomimetic

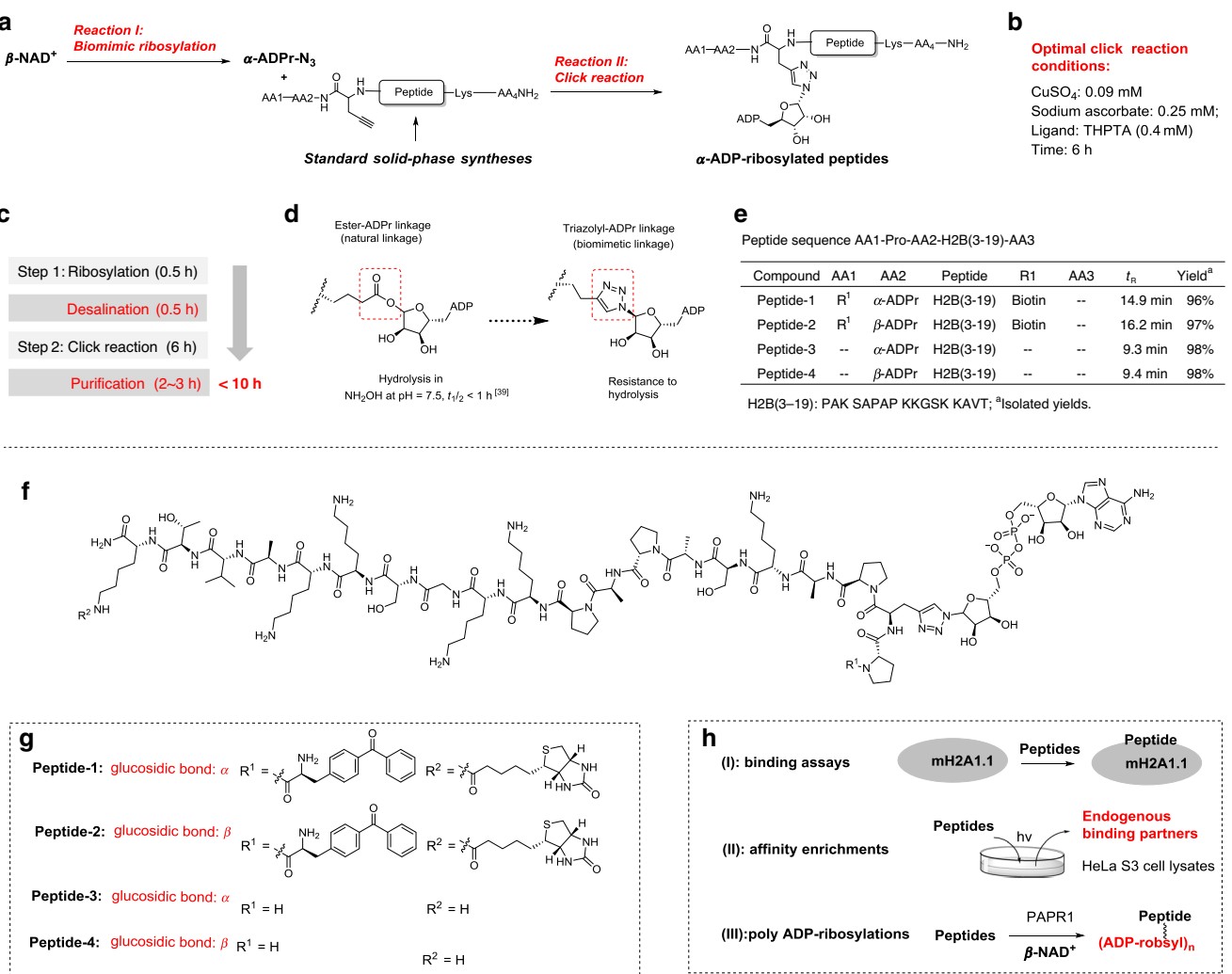

**Fig. 3 Modular synthesis of α-type ADP-ribosylated peptides in two steps for functional investigations. a** The concise synthetic route involving two reaction steps for the preparation of ADP-ribosylated peptides. **b** The optimized catalytic system for the click reaction of α-ADPr-N₃. **c** The programmable steps for preparations of ADP-ribosylated peptides. **d** Replacing the highly labile ester-ADPr linkage with the triazolyl-ADPr linkage. **e** Results of the synthesis of ADP-ribosylated peptides. **f** The chemical structure of peptide **1–4**. **g** The R substituent groups in peptide **1–4**. **h** Biological assays to determine the activities of peptide **1–4**.

ribosylation reactions that can also be performed in open air. Thus, our method for synthesizing ADPr derivatives represents both a convenient and easy approach.

**Modular synthesis of α-type ADP-ribosylated peptides.** De novo synthesis of ADP-ribosylated peptides requires multi-step synthetic routes involving harsh reactions that have low yields (Fig. 1c). Furthermore, unexpected racemizations during the acidic deprotection steps might produce by-products that can be difficult to separate[11]. Therefore, a concise method for the synthesis of ADP-ribosylated peptides is highly appreciated. After using the biomimetic method to obtain α-ADPr-N₃, we further investigated it as a desirable donor of α-ADPr motif for the modular synthesis of α-type ADP-ribosylated peptides, by utilizing the bioorthogonal property of azide group and following click chemistry to connect the peptides bearing alkynyl groups (Fig. 3a). Usually, less than 0.1 mM of copper (II) is suggested for the click chemistry of bioma-cromolecules to ensure good biocompatibility[34]. However, high concentrations of CuSO₄ (10 mM or more) are needed for the click reaction of substrates bearing ADPr units because of binding effects between pyrophosphates and Cu(II)[12,35]. To address this problem, we screened different ligands to improve the catalytic

efficiency of the click reaction for α-ADPr-N₃ (Supplementary Tables 5, 6 in Supplementary Information). We found that tris-hydroxypropyltriazolylmethylamine (THPTA)[29] was an effective ligand for this click reaction, which proceeded smoothly using 0.09 mM of CuSO₄ to give a 95% yield for the ADP-ribosylated product (Fig. 3b and Supplementary Table 6 in Supplementary Information).

After finding an effective click catalyst, we then attempted to use the two-step reaction route to synthesize α-type ADP-ribosylated peptides (Fig. 3c). In the first reaction step, α-ADPr-N₃ was prepared from β-NAD⁺ and sodium azide as starting materials in the [DEOA][Lactate] IL/water system in EP tubes in a heated water bath, which was completed within 30 min. Next, excess sodium azide was removed by a desalination column to avoid adverse effects in the subsequent click reactions. In the second click reaction step, the filtrate including α-ADPr-N₃ was directly reacted with free peptides bearing terminal alkynes in EP tubes in open air, which was completed within 6 h. These free peptides can either be obtained commercially or can be prepared by standard solid-phase synthetic methods. The synthesized α-type ADP-ribosylated peptides were purified by the semipre-parative HPLC using a C-18 silica column using a linear gradient

of 0–50% $CH_3CN$ in 0.1% TFA aqueous solution to give ADP-ribosylated peptide. This integrated synthetic route for α-type ADP-ribosylated peptides requires only two reaction steps in a clean aqueous catalytic system, and all modular operations can be completed within 1 day without the need for specialized equipment or specialist training.

Proof-of-concept applications of the synthetic route were shown in Fig. 3d–g. A histone H2B peptide with an ADP-ribosylation mark at the Glu2 position, a reported endogenously ADP-ribosylated site[36–38] was chosen as the model. The intrinsic chemical instability of Glu-ADP-ribosylation makes it difficult to study the effects of ADP-ribosylation at this site[4,39–43]; therefore replacing the highly labile ester-ADPr linkage in the native H2B peptides with a triazolyl-ADPr linkages might produce a stable chemical mimic peptide that could facilitate the investigation of its precise biological interactions (Fig. 3d). For the peptide probe design, we appended a photoreactive group, benzophenone, at the N-terminus of the peptide to covalently capture its binding partners upon UV irradiation, whereas biotin was used to tag the C-terminal lysine residue for the detection and enrichment of captured proteins (Fig. 3f, g). To introduce the triazolyl-ADPr linkage, the original Glu2 site on H2B peptide was replaced by propargylglycine, an alkyne-containing amino acid, during solid-phase synthesis. To illustrate the critical roles of the α-type ADPr glycosidic bond, peptides with β-ADP-ribosylation marks (Peptide-2 and Peptide-4 in Fig. 3f, g) were used as the controls. All the designed peptides were synthesized efficiently using our two-step modular route with the isolated yields over 96% (Fig. 3e and Supplementary Table 7, Fig. 7 in the Supplementary Information). In contrast to the natural ester-linked ADP-ribosylated peptides that are unstable in $NH_2OH$ treatment at neutral pH ($t_{1/2} < 1$ h)[39,41,42], all the peptides bearing triazolyl-ADPr linkages were stable for at least 24 h in $NH_2OH$, as well as in acidic and basic conditions (Supplementary Fig. 8 in the Supplementary Information). This desirable chemical stability then allowed us to investigate their bioactivities in the three groups of assays (Fig. 3h)

**Affinity binding assay with macroH2A1.1.** We first determined whether the designed peptides (Fig. 3f) could mimic the interactions of the native ADP-ribosylated histone in binding to the ADP-ribosylation-dependent proteins. We adopted the well-characterized recognition of protein ADP-ribosylation by macrodomains. MacroH2A1.1 (mH2A1.1) was incubated with a series of concentrations of peptide-1 before being subjected to UV irradiation. The resulting mixture was separated by gel-electrophoresis and detected by streptavidin blotting. We observed the dose-dependent labeling of mH2A1.1 by peptide-1 ($EC_{50} = 2.1 \mu M$) (Fig. 4a), which was similar to the reported dissociation constant between the macrodomain of MacroD2 and the ADP-ribosylated H2B peptide that has the same sequence and modification site[11]. On the contrary, labeling of mH2A1.1 by control peptide-2 that has a β-triazolyl-linked ADPr was significantly weaker and inefficient as expected (Fig. 4b). In addition, labeling by peptide-1 was blocked in the presence of peptide-3 ($IC_{50} = 4.0 \mu M$, Fig. 4c), whereas the β-epimer peptide-4 hardly competed with the cross-linking (Fig. 4d), further demonstrating that the α-glucosidic bond is critical for the recognition of ADP-ribosylation by mH2A1.1. We also showed that peptide-1, at the required concentration for robust labeling of mH2A1.1, failed to capture either the mH2A1.1 with a G224E mutation, an ADPr-binding-deficient mutant, or capture mH2A1.2, a nonADPr-recognizing isoform of mH2A1.1 (Fig. 4e). These results showed the designed α-ADP-ribosylated peptides had excellent performance in mimicking their native prototypes with high affinities. More importantly, the stable triazolyl linkage offers a class of

useful chemical tools with controllable stereochemistry at the glutamate-type glycosidic bond, which can be used to study the functional differences between the α-ADP-ribosylation and β-ADP-ribosylation epimers[44].

**Enrichment of binding partners in cellular lysate.** Encouraged by the results obtained using single proteins, we next explored the application of the prepared α-ADP-ribosylated peptides in complex proteomes to determine if the endogenous binding partners of ADP-ribosylation can be captured. To this end, peptide-1 was incubated with lysates derived from HeLa S3 cells, followed by photo-cross-linking and streptavidin enrichment. The resulting protein mixtures were analyzed by immunoblotting using anti-bodies against two known endogenous ADP-ribosylation binding proteins, mH2A1.1 and PARP9 (Experimental details for immunoblotting is detailed on Page 67 in the Supplementary Information). The result showed that peptide-1 was able to enrich both proteins in the HeLa S3 cell lysate (Fig. 4f). Moreover, pretreatment of the sample with peptide-3 largely reduced the enrichment efficiency. This result shows the potential of using such peptides for the study of ADP-ribosylation-mediated protein–protein interactions in physiologically relevant contexts, and identification of previously unknown binding partners of ADP-ribosylation.

**Poly ADP-ribosylation (PARylation) catalyzed by PARP1.** PARylation involves the incorporation of multiple ADPr units after the installation of the first ADPr unit on the protein substrates[1,45–47]. Poly ADPr on glutamic acid have biological functions[38,48–50], which inspired us to test whether our α-type ADP-ribosylated peptides could be used to nucleate the site-specific generation of poly-ADP-ribosylated peptides. Peptide-3 with an α-triazolyl-linked ADP-ribosylation was used in the poly-ribosylation reaction with poly(ADP-ribose) polymerase 1 (PARP1) and biotinylated β-$NAD^+$. After the reaction, the mixture was monitored by streptavidin blotting, which showed a clear biotin signal (Fig. 4g). No biotin signals were observed in samples without added PARP1 and/or biotinylated β-$NAD^+$, or with unmodified histone H2B peptide (with a glutamate at the 2nd position). These results indicated that peptide-3 could be transferred with more ADPr units by PARP1, but peptides without triazolyl-linked ADP-ribosylation did not lead to PARylation due to the lack of ADPr templates. To further monitor the growth of poly-ADP-ribose chain, we performed the poly ADP-ribosylation reaction using the biotin-tagged peptide-1 and natural β-$NAD^+$. The reactions were carried out with different incubation times or with different concentrations of β-$NAD^+$. The resulting mixtures were subjected to streptavidin enrichment prior to immunoblotting analysis against an anti-poly-(ADP-ribose) polymer antibody to avoid the signal interference of auto-poly ADP-ribosylation of PARP1. The results (Supplementary Fig. 10 in the Supplementary Information) showed that, with the increase of the reaction time or the concentration of β-$NAD^+$, the region with robust poly ADP-ribosylation signal gradually expanded throughout the whole lane with increasing intensity, suggesting the generation and growth of poly ADP-ribose chain on peptide-1 in both a time-dependent and a β-$NAD^+$ concentration-dependent manners. More specifically, the presence of poly-ADP-ribose signals at high molecular regions clearly indicated that even hundreds of ADP-ribose units could be added to the peptide by PARP1. It was recently reported that, besides the Glu 2, other amino acid residues in the histone H2B N-terminal tail could also be poly-ADP-ribosylated by PARP enzymes, among which the Ser 7 and Ser 15 residues are two major modification sites[51]. To determine whether the

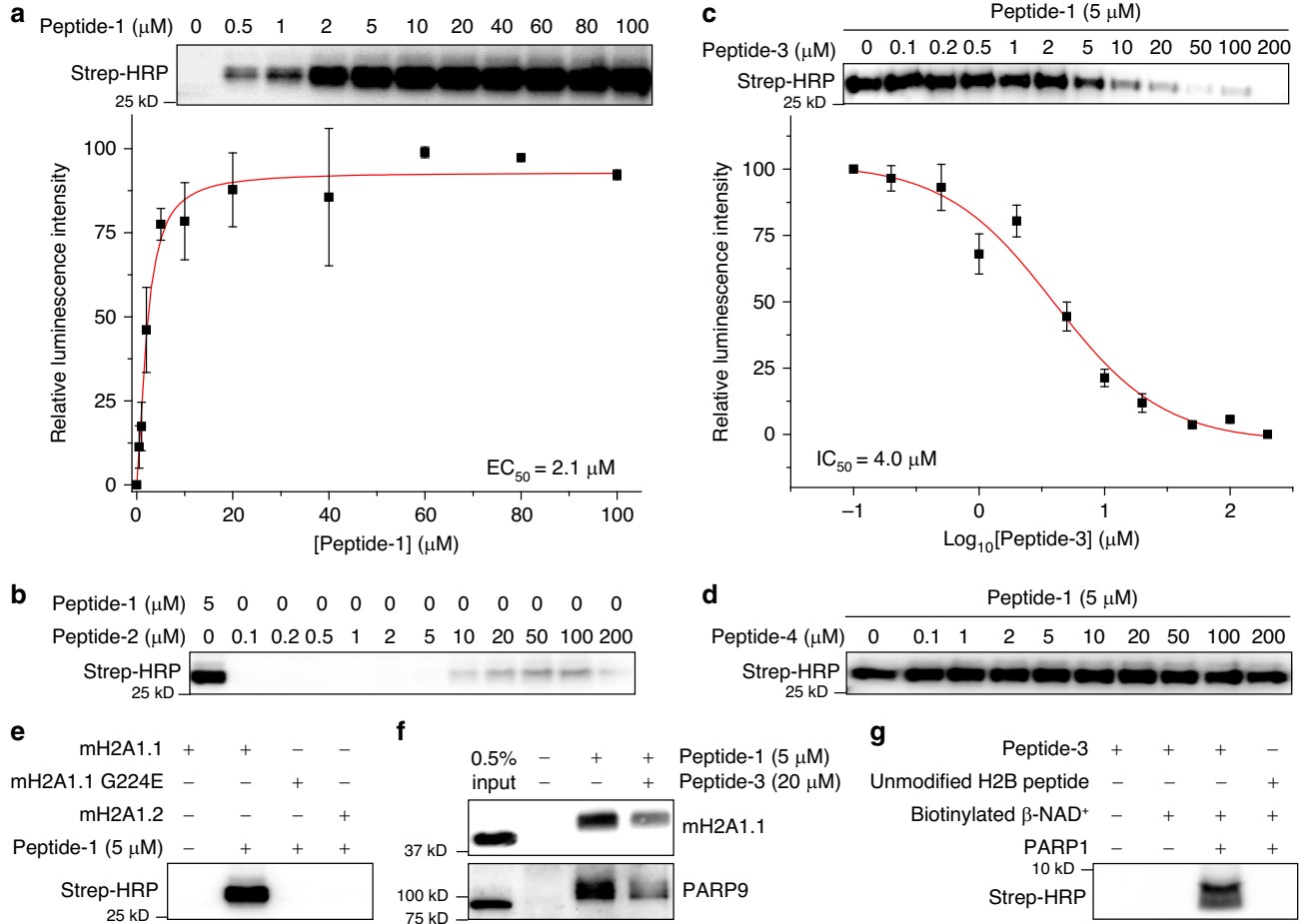

**Fig. 4 The triazolyl-linked ADP-ribosylation represents a good mimic of the natural ADP-ribosylation. a** α-ADP-ribosylated peptide-1 labeled mH2A1.1 in a concentration-dependent manner. Data are reported as mean ± s.d. (*n* = 3). **b** β-ADP-ribosylated peptide-2 labeling of mH2A1.1 was much weaker and inefficient compared with peptide-1. **c** Competition between α-ADP-ribosylated peptide-1 and peptide-3 in labeling mH2A1.1. Data are reported as mean ± s.d. (*n* = 3). **d** β-ADP-ribosylated peptide-4 failed to compete with peptide-1 in labeling mH2A1.1. **e** Peptide-1 selectively labeled mH2A1.1 but not mH2A1.1 G224E mutant or mH2A1.2. **f** Peptide-1 enriched endogenous mH2A1.1 and PARP9 from HeLa S3 cell lysate. **g** The α-ADP-ribosylated peptide-3 serves as a substrate of PARP-1-induced poly ADP-ribosylation. In **a**, **c**, **e**, **f**, and **g**, the blottings represent three independent experiments. In **b** and **d**, the blottings represent two independent experiments. In **a** and **c**, curves were normalized between 100% and 0% at the highest and lowest luminescence. See "Photo-cross-linking and visualization of the biotinylated proteins" in the "Methods" section for more details of the photo-cross-linking-based assays. The uncropped blots and protein markers are provided in Supplementary Fig. 9 in the Supplementary Information. Source data of uncropped versions of gels or blots in Fig. 4 are provided as a Source Data file (https://doi.org/10.6084/m9.figshare.12375032).

poly-ADP-ribosylation catalyzed by PARP1 occurs at the pre-existing triazolyl-ADPr mark or at other Ser residues, we synthesized a new peptide-1 analog (peptide-5, Supplementary Fig. 11a in the Supplementary Information), in which the Ser 7 and Ser 15 were mutated to Ala residues that could not be decorated by (poly-)ADP-ribosylation. We next repeated the poly-ADP-ribosylation reaction using both peptide-1 and peptide-5 in parallel. The resulting mixtures were subjected to immunoblotting analysis using the anti-poly(ADP-ribose) polymer antibody. The result (Supplementary Fig. 11b in the Supplementary Information) showed that both peptides offered comparable levels of overall chemiluminescence signals, suggesting that the two peptides were decorated by similar extents of poly-ADP-ribosylation under the same conditions. Although this observation could not entirely rule out the possibility that the poly-ADP-ribosylation also took place on the Ser 7 and/or Ser 15 residues of the H2B peptide, it did support that the preexisting triazolyl-ADPr mark was preferred by PARP1 to put on extra ADP-ribose units as the dual Ser-to-Ala mutations did not lead to significant differences. These findings on poly ADP-ribosylation

are expected to expand the applications of our method, allowing the preparation of more complicated poly-ADP-ribosylated peptides for the study of PARylation, the determination of mechanisms regulating PARP, as well as the development of antibodies for site-specific protein poly-ADP-ribosylations[8].

## Discussion

In this work, we developed an alternative approach to construct α-type ADP-ribosylated peptides. Using a biomimetic ADP-ribosylation reaction with β-NAD+ as the ADPr donor, key intermediates (α-ADPr-N₃) can be prepared in simple [DEOA][Lactic] IL/water systems with 25:1 α/β selectivity and yields of around 86%. This reaction in tandem with click chemistry then offers a modular synthesis of α-type ADP-ribosylated peptides. The concise synthetic route provides a fast access to biologically important ADP-ribosylated peptides with controllable sequences and functional modifications. Using the human histone H2B peptide as a model, we demonstrated the prepared α-type ADP-ribosylated H2B peptides not only have high affinities with intact

ADP-ribosylation binding domains at the molecular level, but also have desirable cellular stability in enriching endogenous ADP-ribosylation protein partners in cell lysates. Furthermore, the peptides could be used as effective substrates to prepare more complicated site-specific poly ADP-ribosylated peptides.

ADP-ribosylation is a highly dynamic and reversible post-translational modification. The increasing importance of ADP-ribosylation in physiological and pathological functions demands well-defined α-type ADP-ribosylated peptides as the chemical tools for studying molecular mechanisms of various ADP-ribosylations, however accessing the peptides is challenging due to their inherent complicated structures and the lack of effective synthetic tools. The syntheses of α-type ADP-ribosylated peptides described in this work therefore supply a practical approach, by which all the reactions can be performed open air in EP tubes without the need for specialized instruments or training. Together with the other advantages, including easily accessible materials, precise stereochemical control and concise synthetic routes, high yields and multidimensional bio-applications of products, the presented methods may become a powerful synthesis platform to produce general molecular tools for the study of protein ADP-ribosylation.

## Methods

**The screening of water/IL systems for the transformations of β-NAD$^+$ to α-ADPr-N$_3$.** 54 kinds of ionic liquids with different anions and cations (Supplementary Table 2) were prepared and identified based the supplementary methods and synthetic routes (Supplementary Fig. 1) in Supplementary Information. Screening of different ionic liquids for the stereoselective transformation of β-NAD$^+$ to α-ADPr-N$_3$ were conducted with the following experimental procedure. Typically, β-NAD$^+$ (0.5 mg, 0.75 μmol), NaN$_3$ (2 mg, 0.03 mmol) in the mixture of H$_2$O and ionic liquids (molar ratio of H$_2$O:IL was 5:1) were put in a PCR instrument at 90 °C for 0.5 h. And then, the reaction mixtures were analyzed by HPLC equipped by C-18 column. Ratio of α/β-ADPr-N$_3$ was calculated by the peak area of α/β-ADPr-N$_3$. Further data analyses of the effects of anions and cations on the ratio of α/β-ADPr-N$_3$ and their yields were shown in Supplementary Figs. 1–6 in the Supplementary Information.

**General procedures for synthesis of ADPr-N$_3$ and derivatives.** The mixture of β-NAD$^+$ or its derivatives (0.075 mmol) and 196 mg of sodium azide (3 mmol) in the mixed solvents (0.5 mL) of H$_2$O and [DEOA][Lactate] (mol/mol = 5:1) was heated at 90 °C. The reaction procedure was detected by HPLC. When the peak of the raw material disappeared or not reduced, the products were purified by preparative HPLC. The column was eluted using a linear gradient of 0–10% CH$_3$CN in TEAB buffer (0.1 M, pH 7.5) within 35 min to give α-type ADPr-N$_3$ and β-type ADPr-N$_3$ with different retention times. Scale-up synthesis of α-ADPr-N$_3$ (**1**) was conducted with 0.5 g of β-NAD$^+$ and 5.0 g sodium azide as the starting materials. The mixture in 5 mL mixed solvent of H$_2$O and [DEOA][Lactate] (mol/mol = 5:1) were stirred at 90 °C for 2 h, and then purified by preparative HPLC a linear gradient of 0–10% CH$_3$CN in TEAB buffer (0.1 M, pH 7.5). The isolated yield was calculated after lyophilization.

**General procedures for modular synthesis of ADP-ribosylated peptides from β-NAD$^+$.** i. Ten milligram of β-NAD$^+$ (0.015 mmol) and 40 mg of sodium azide (0.6 mmol) were added into a 0.5 mL EP tube containing in the mixed solvents (0.1 mL) of H$_2$O and [DEOA][Lactate] (mol/mol = 5:1). The resulted EP tube was heated at 90 °C for 0.5 h, and then cooled to room temperature. ii. The reaction mixture was desalinated by a short C-18 column with water and then 50% CH$_3$CN in water. iii. Alkynyl-containing polypeptide (0.012 mmol) were dissolved into the concentrated desalinated solution of α-ADPr-N$_3$ (around 8.0 mg, 0.014 mmol) with copper sulfate pentahydrate (0.09 mM), THPTA (0.36 mM) and sodium ascorbate (0.25 mM) in a total 1.0 mL of water. The mixture of click reaction was kept at 37 °C for 2–4 h. iv. Separation and purification of ADP-ribosylated peptides were conducted using preparative HPLC with a linear gradient of 0–50% CH$_3$CN in 0.1% TFA aqueous solution. The target products were obtained as white powders after lyophilization.

**Photo-cross-linking and visualization of the biotinylated proteins.** The photoaffinity peptide probe (peptide-1 or peptide-2) at indicated concentration was incubated with recombinant proteins (20 ng/μL) or cell lysates (1 mg/mL) in binding buffer (50 mM HEPES, 150 mM NaCl, 2 mM MgCl$_2$, 0.1% Tween-20, 20% glycerol, and pH 7.5) for 10 min on ice. For competition samples, indicated competitor (peptide-3 or peptide-4) were preincubated with the proteins or lysates for 15 min on ice before the addition of photoaffinity peptide probe. The samples were then irradiated at 365 nm for 20 min on ice.

The proteins after photo-cross-linking were precipitated by adding 5 volumes of ice-cold acetone and placed at −20 °C overnight. After protein precipitation, samples were centrifuged at 6000 × $g$ for 5 min at 4 °C. The supernatant was discarded, and the pellet was washed with ice-cooled methanol twice and air-dried for 10 min. The proteins were resuspended in 1× LDS sample loading buffer (Invitrogen) with 50 mM DTT and subjected to immunoblotting detection.

**Enrichment of the biotinylated proteins.** The proteins after photo-cross-linking were precipitated by adding 5 volumes of ice-cold acetone and placed at −20 °C overnight. After protein precipitation, samples were centrifuged at 6000 × $g$ for 5 min at 4 °C. The supernatant was discarded, and the pellet was washed with ice-cooled methanol twice and air-dried for 10 min. The protein pellet was then dissolved in 1× PBS with 4% SDS, 20 mM EDTA, and 10% glycerol by vortexing and heating. The solution was then diluted with 1× PBS to give a final concentration of SDS as 0.5%. High capacity streptavidin agarose beads (Thermo Fisher Scientific) were added to bind the biotinylated proteins with rotating for 2 h at room temperature. The beads were washed stepwise with 1× PBS with 0.2% SDS, 6 M urea in PBS with 0.1% SDS, and 250 mM NH$_4$HCO$_3$ with 0.05% SDS. The enriched proteins were then eluted by heating in 1× LDS sample loading buffer (Invitrogen) with 50 mM DTT and subjected to immunoblotting detection.

**Poly ADP-ribosylation at peptide level by PARP1.** i. For experiment using peptide-3 and biotinylated β-NAD$^+$. To peptide-3 or unmodified H2B peptide (15 μM) in reaction buffer (50 mM Tris pH 8.0, 10 mM MgCl$_2$, 1 mM DTT) containing 1× activated DNA (Trevigen) was added biotinylated NAD$^+$ (10 μM) or/and PARP1 (1 μL, 10 units/μL, Trevigen) as indicated. Reactions were performed in 20 μL total volume at 25 °C for 1 h. Reactions were stopped with 7 μL 4× LDS sample loading buffer (Invitrogen), and subjected to immunoblotting. ii. For experiment using peptide-1 or peptide-5 and natural β-NAD$^+$. To the peptide (15 μM) in reaction buffer (50 mM Tris pH 8.0, 10 mM MgCl$_2$, 1 mM DTT) containing 1× activated DNA (Trevigen) was added indicated concentration of β-NAD$^+$ and PARP1 (1 μL, 10 units/μL, Trevigen). Reactions were performed in 20 μL total volume at 25 °C for indicated period of time. Reactions were stopped by adding 1 μL 20% SDS and heating at 85 °C for 5 min. Eighty microliter of PBS was added to reduce the SDS concentration to 0.2%. Two milligram prepared Dynabeads (Dynabeads preparation: 2 mg Dynabeads was washed by 1 mL PBS with 0.1% Tween-20 for five times, followed by 1 mL PBS with 0.2% SDS for five times) suspended in 400 μL PBS containing 0.2% SDS were added to the mixture, which was allowed to incubate for 2 h at room temperature by rotation. The resulting supernatant was discarded. The beads were washed with 1 mL PBS containing 0.2% SDS for five times. To the collected beads, 20 μL 1× LDS sample loading buffer (Invitrogen) was added and boiled at 95 °C for 10 min, followed by immunoblotting.

**Reporting summary**. Further information on research design is available in the Nature Research Reporting Summary linked to this article.

## Data availability

The data supporting the findings of this work are available within the article and its Supplementary Information files. Source data underlying Fig. 2d and Supplementary Figs. 2–7 and uncropped versions of gels and blots presented in the Fig. 4 are available in figshare (https://doi.org/10.6084/m9.figshare.12375032). All data are available from the authors upon reasonable request. Source data are provided with this paper.

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

## Acknowledgements

This work was funded by the National Natural Science Foundation of China (21778015, 21373079, and U1704251 to L.L.; 91753130 to X.D.L.), Excellent Young Scientists Fund of China (Hong Kong and Macau) (21922708 to X.D.L.), the 111 Project (No. D17007 to L.L.) by the Chinese Ministry of Education. We acknowledge the support from the Hong Kong Research Grants Council (RGC) Collaborative Research Fund (CRF C7029-15G and C7017-18G to X.D.L.), the Areas of Excellence Scheme (AoE/P-705/16 to X.D.L.), the General Research Fund (GRF 17121120, 17126618, and 17125917 to X.D.L.), and the RGC Postdoctoral Fellowship (Scheme 2020/21 to X.L.). G.Z. acknowledges the support from Henan Normal University Doctoral Initiation Fund (qd18018).

## Author contributions

L.L., A.Z., and L.Z. design the chemical experiments. X.L. and X.D.L. designed the biological experiments and analyzed the data. A.Z., B.L., and G.Z. performed the preparations of ionic liquid pool and synthesis of target compounds. A.Z., Y.G., and J.W. analyzed the screening data. J.L., Y.C., and G.T. constructed plasmids, expressed, and purified the proteins used in this study. X.L. performed the photo-cross-linking and the poly-ADP-ribosylation assays. X.L., A.Z., X.D.L., and L.L. wrote the manuscript.

## Competing interests

The authors declare no competing interests.
