## [Peer Review File · Nature Communications]

Reviewers' comments:

Reviewer #1 (Remarks to the Author):

The paper "Biomimetic α -selective ribosylations (sic) enable two-step modular synthesis of biologically important ADP-ribosylated peptides", by Li et al. describes a synthetic approach to generate functionalised NAD derivatives, cleanly with high stereoselectivity and in high yield. These derivatives are then utilised with copper-catalysed click chemistry to produce modified, ADP-ribosylated peptides, whose activities in pull-down assays in vitro and in lysate validate their binding properties and indicate future utility in biochemical studies of these functionally important peptide derivatives. A key stage in the synthesis has been to utilise ionic liquids through a newly-developed high-throughput screening process (Green Chemistry, 2019) to enhance the reactivity and stereoselectivity of the initial reaction with excess sodium azide, which is also shown to translate to reaction with bromine to generate the corresponding bromide. The rapid, bench-top synthesis, which was shown to scale and use accessible materials, will be of significant interest to researchers wishing to study the properties and roles of this important functional modification (in signal transduction, cellular differentiation, gene regulation, cancer, and other topics of general interest) and, on this basis, is likely to make the results of this paper impactful and timely. The highlighting of the screening approach will also help in directing future method development in a number of synthetic fields.

There are a number of questions arising from the experiments, as presented in the text, that may help in greater understanding of the rationale, in addition to modifications to demonstrate full characterisation and improve clarity.

- 1) A range of ionic liquids have been used, and have clearly in some cases shown good results. However, it is not clear, since these are run as aqueous solutions, whether this is a simple (chiral) ion effect. Clearly lactate (cf. lactic in main text) is important – how do other lactate salts fare, and are ionic liquids per-se required?
- 2) The choice of 5:1 mole fraction water:IL is selected based on one set of screening results, and not the IL that was finally recommended. How reasonable is this selection and why weren't other ILs of interest tested in this framework? Or, rephrased, how likely are the results from these ratios to be comparable across different IL systems.
- 3) The highlighted ionic liquids in Figure 2D that are indicated as being taken forward to screening are not necessarily those that show the best ratios? What governed this selection? Why were other reasonable candidates excluded ahead of those showing poorer ratios? How were the error-bars calculated (numbers of replicates are not clearly highlighted for example in the captions)?
- 4) BF₄ anion was screened as part of the set. Why was this, given that this anion is recognised to undergo hydrolysis in aqueous solutions? Further, these anions do not appear in the list in Figure 2B?
- 5) Is there scope for a 1-pot process utilising the ionic liquid to aid the click reaction?
- 6) How were the optimisation parameters/ranges selected? From the SI this seems to have been an improvement process rather than optimisation.
- 7) How was the physiological stability of the ribosylated peptides measured and under what conditions? These data should be included in the SI.
- 8) For the ionic liquids: the anion names should be IUPAC consistent (-oate for carboxylates). The ¹H NMR data look very good, but I would normally expect both a reference to literature data for each of the synthesised materials and, for known compounds, 2 pieces of spectral and 1 piece of physical data to confirm identity and purity. This should be the case for all characterised compounds, unless there are clearly justified reasons as to why these cannot be obtained.
- 9) In the case of the α - (1) and β - (1') azides, the ¹H NMR clearly indicates impurities/isomers/residual starting material. Please indicate further what these peaks are and their impact.
- 10) Figures S7 and S9 it would help to mark the relevant cross-peaks on the spectra for clarity.
- 11) For all figures in the SI, it would be helpful to have more detailed captions.
- 12) Figure S11 could be improved by angling the triazole in each case so that it doesn't clash with the proline.

13) For the synthesis of ribosylated proteins the time is quoted for the procedure as 2-4h. This should be more precisely identified; whether this is a per peptide difference, or some other reason. The MALDI data should have all peaks assigned.

14) It would be helpful to link the immunoblots in Figure 4 to the experimental in the SI. Although this can be worked out (the captions are fine), it would improve the clarity for the reader.

15) Overall the manuscript would benefit from a thorough language editing. Currently some aspects are either hard to understand or not well specified and there are a number of typographical errors. This is less of a problem for the SI, where there are only a few minor issues, but these should also be looked at.

Review for NCOMMS-19-34957-T

The manuscript by Zhu et al. describes a new synthetic strategy for the generation of an azido-ADP-ribose analog (N_3 -ADPr). This analog contains an azide group at the 1' position of the glycosidic ring. The synthesis uses ionic liquids and the authors developed an efficient synthesis for generating the biologically relevant alpha isomer of N_3 -ADPr. They then coupled N_3 -ADPr to an H2B peptide (known ADP-ribosylation target of PARP1). The peptide also contains a benzophenone crosslinker as well as a biotin affinity handle. The authors showed that the peptide could bind to a macro domain protein mH2A1.1, which is a protein known to recognize ADP-ribose. The authors also showed that this peptide could pulldown known ADP-ribose binding macro domain containing proteins (i.e. mH2A1.1 and PARP9) from HeLa cell lysates that were treated with UV light to induce crosslinking. Lastly, the authors showed that the ADPr-peptide could be ADP-ribosylated by PARP1.

Overall this is an interesting study; however, there are some major issues that need to be addressed prior to publication. These include:

1. Overall, the manuscript really needs to be carefully edited. There are many grammar and spelling mistakes. Additionally there are issues regarding scholarship (more on this below).
2. In the paper, the authors state that the half-life for Glu/Asp-ADPr is < 1 h. They ref. 31, but this ref. doesn't state this. Please provide the correct ref. for this statement or remove.
3. In the manuscript text, the IC_{50} values stated in the text are molar but they should be micromolar according to Fig. 3. Also, the authors that the values they obtained are similar to literature values, but different macro domain proteins were tested (i.e. MacroD2 and TARG1 catalytic dead mutants in the ref: "Synthetic α - and β -Ser-ADP-ribosylated Peptides Reveal α -Ser-ADPr as the Native Epimer," *Organic Letters* 40, 4140-4143 (2018). Also the position of the ADPr is different-they should comment on this.
4. One major issue is that the ADPr peptide they synthesized has the triazol-linked ADPr in a non-native position. The authors should synthesize a peptide that places the ADPr at the serine (Ser10 in the KS motif) that gets ADPr by PARP1, similar to what was done in *Organic Letters* 40, 4140-4143 (2018).
5. The manuscript states: "And more importantly, the stable triazol linkage offered, to the best of our knowledge, the first class of chemical tools with controllable stereochemistry at the glutamate-type glycosidic bond to study the functional differences between the α - and β -ADP-ribosylation epimers." This is not accurate. See this ref: "Synthetic α - and β -Ser-ADP-ribosylated Peptides Reveal α -Ser-ADPr as the Native Epimer," *Organic Letters* 40, 4140-4143 (2018). The authors should discuss this paper.
6. Fig 1B: replace asparaginic with aspartate for consistency with glutamate.
7. Fig 1: the authors should also include the previous synthesis of N_3 -ADPr (*Molecules*, 2017) in Fig. 1.
8. Fig. 4: are there controls for crosslinking? Also, in Fig. 4f, the last lane should be "+" for peptide-3 competition if I understand the experiment correctly.
9. Fig. 4: in the main text, the wrong panel is referenced in several sections.
10. Fig 4f: Did the authors consider doing MS/MS proteomics? In this way, they could discover potentially new ADPr binding proteins. This would certainly add novelty to the manuscript as this has not been done before with other ADPr-peptides.
11. Fig. 4: Should include molecular weights next to blots.

12. Fig. 4g: see point 4. The correct peptide should be used before drawing any conclusions about the ADPr serving as a “priming” site. Additionally, the authors should use native NAD⁺ and probe using commercially available poly-ADPr antibodies to show that poly-ADP-ribosylation occurs. And the authors need to determine if it is happening on the triazol-linked ADPr position and not another amino acid in the peptide.
13. This statement needs references: “especially for those on glutamate residues, can be removed by multiple enzymes,”...
14. Supplemental data: The ¹H NMR for the alpha and beta N₃-ADPr contain some of the other isomer.

Reviewer #3 (Remarks to the Author):

The manuscript by Li and co-workers describes the facile synthesis of an azido-analog of NAD, followed by the use of this compound to create triazole linked ADP-ribosylated peptide analogs for further biological studies. They then apply these peptides to demonstrate the enrichment of proteins that are known to bind to an ADP-ribosylated H2B peptide in vitro and from cell lysates. I feel that this approach is attractive and that the conclusions are mostly supported by the data. In particular, the synthesis of the azido-analog of NAD, and other nucleotides, using ionic liquids is in particular quite unique and powerful. I also agree with the authors that the procedure seems easy enough, with minimal purification steps, to be performed by many labs that could then go on to use the modified peptides. However, I do have some comments that should be addressed in a revised manuscript.

Comment #1: Overall the paper is difficult to read due to numerous grammatical errors that need to be addressed.

Comment #2: How was the stability of the ADP-ribosylation triazole-analogs of the peptides measured? I may have simply missed it, but this data should be added to the Supplementary Information.

Comment #3: The ordering and "call-outs" for most of the panels in Figure 4 seem to be confused at places in the text and the figure legend.

Comment #4: In Figure 4F, the "truth-table" indicates that peptide 3 is missing from all of the conditions. This should be fixed.

Comment #6: The authors should test the poly-ribosylation of their analog further. They show in current Figure 4G that PARP1 can modify peptide 3 with biotinylated NAD. However, they do not appear to show how extensive this polymerization is. They should do this by labeling peptide 1 with regular NAD⁺ and showing the full molecular weight spread of the product by blotting for the biotin on peptide 1. This will give an idea of how long the poly-ADP ribosylation can become.

Comment #7: As a last characterization experiment, the authors should consider using peptide 1 to perform an unbiased proteomics experiment to demonstrate the utility of the peptides as a discovery-based technology.

Point-by-Point Responses to the Reviewers' comments

We greatly appreciate the comments of the reviewers on our manuscript. Based on these comments, we have performed a series of supplementary experiments. All the unreported 47 ionic liquids were characterized by ^{13}C NMR and IR besides of ^1H NMR. The stability of our ribosylated peptides were measured in 0.5 M NH_2OH (pH = 7.5) and conditions with pH = 3.0, 5.0, 9.0 and 11.0 respectively. The growth of a poly-ADP-ribose chain was indicated to contain at least 35 ADP units using native $\beta\text{-NAD}^+$ and peptide-1 as the substrates. Additional control experiments were conducted to explain the effects of ionic liquids on reaction selectivity, which suggested both ion environments and synergism of cation and anions play important roles in the current biomimetic reaction. Besides, the complete MALDI-TOF-MS spectra, clean ^1H NMR spectra of $\alpha\text{-ADPr-N}_3$ (1) and $\beta\text{-ADPr-N}_3$ (1'), and the uncropped membranes together with protein markers for all the blots in Fig. 4 in manuscript were added into the revised supporting information.

We have also reorganized the context that reviewers concerned in the manuscript to clarify the novelty of this work. We checked the typographical errors carefully in manuscript and supporting information, revised the abstract to 150 words and added the part of "Methods" in the manuscript based on requirements of journal, and the specialized language editing was conducted for the revised manuscript. The supporting information was reformulated with the revised figures and supplementary experiments. We believe that the revised manuscript has been substantially improved from the revisions and additional new data. All the revised sentences were marked with **yellow** in the revised manuscript. The details are as follows.

Part-I: Response to reviewer#1

The paper "Biomimetic α -selective ribosylations enable two-step modular synthesis of biologically important ADP-ribosylated peptides", by Li et al. describes a synthetic approach to generate functionalized NAD derivatives, cleanly with high stereoselectivity and in high yield. These derivatives are then utilized with copper-catalyzed click chemistry to produced modified, ADP-ribosylated peptides, whose activities in pull-down assays in vitro and in lysate validate their

binding properties and indicate future utility in biochemical studies of these functionally important peptide derivatives. A key stage in the synthesis has been to utilize ionic liquids through a newly-developed high-throughput screening process (Green Chemistry, 2019) to enhance the reactivity and stereoselectivity of the initial reaction with excess sodium azide, which is also shown to translate to reaction with bromine to generate the corresponding bromide. The rapid, bench-top synthesis, which was shown to scale and use accessible materials, will be of significant interest to researchers wishing to study the properties and roles of this important functional modification (in signal transduction, cellular differentiation, gene regulation, cancer, and other topics of general interest) and, on this basis, is likely to make the results of this paper impactful and timely. The highlighting of the screening approach will also help in directing future method development in a number of synthetic fields.

There are a number of questions arising from the experiments, as presented in the text, that may help in greater understanding of the rationale, in addition to modifications to demonstrate full characterization and improve clarity.

Question-1: A range of ionic liquids have been used, and have clearly in some cases shown good results. However, it is not clear, since these are run as aqueous solutions, whether this is a simple (chiral) ion effect. Clearly lactate (cf. lactic in main text) is important – how do other lactate salts fare, and are ionic liquids per-se required?

Response: We thank the reviewer for this comment. It is very interesting to study how the ionic liquid aqueous system affect the reaction (transfer β -NAD to α -ADP-ribose). We first investigated the ion effects. Inorganic salts including NaCl, KCl and CaCl₂ were used to replace the ionic liquids for the reaction. The results were shown in Table R1 below. To our surprise, a dramatic salt effect was found in which yield of α -ADPr-N₃ could be obtained in 55.3% with NaCl aqueous solution (Entry 2 in Table R2), which was over 6 times than 8.5% that was obtained under the same conditions in the absence of NaCl (Entry 1 in Table R1). The similar result was obtained with KCl aqueous solution with a 53.7% yield. Then we tried lactate salts including sodium lactate, potassium lactate and calcium lactate. In the solutions of sodium lactate and potassium lactate, high yields could be also obtained with 58.5% and 59.2% respectively. **Therefore, both inorganic and organic salts in aqueous solution could effectively promote reactions to produce α -ADPr-N₃, suggesting ion environments might play a role during this reaction. Besides, the**

type of cations in the salts affects the reaction output. For example, calcium salts such as CaCl_2 and calcium lactate gave reduced yields (22.7% and 23.6% respectively).

The high α -selectivity and yield (reactivity) in the current reactions derive from synergistic effects of cations and anions of ionic liquids. Firstly, we compared the ionic liquids that all contained lactate as the anions. As shown in Table R2, only [DEOA][Lactate] and [DMEOA][Lactate] can give the ratio of α/β more than 20.0, other cations like choline, TEOA and TMG gave the ratio of α/β from 10.7 to 14.9, and even lower with DBU as the cations (7.8 in Entry 1 in Table R2). Secondly, we compared the ionic liquids that all had DMEOA as the cations. As shown in Table R3, all the ionic liquids could promote the reactions to give α -ADPr- N_3 with high yields over 44%, however showed the dramatic differences in selectivity. Lactate represented the anion to pair with DMEOA to give the highest ratio of α/β (24.7), while hexanoate represented the anion to pair with DMEOA to give the lowest ratio α/β (2.1). Thirdly, we analyzed the distribution of the cations and anions in liquid liquids that can promote the reaction with high selectivity (ratio of α/β more than 10.0) and high yield (yields more than 40.0%). As shown in Table R4 below, 6 cations and 4 anions are included (cations: Choline, DEOA, TEOA, DMEOA, TMG, BMIM; anions: Lactate, Glycolate, Acetate, BF_4) to give the ratio of α/β more than 10.0. As to the ionic liquids with high yields over 40%, 4 cations and 7 anions were included as shown in Table R5 below (cations: Choline, TEOA, DEOA, HMIM; anions: Benzoate, Glycolate, Hexanoate, Butyrate, Acetate, Lactate, BF_4). Indeed, higher yields of α -ADPr- N_3 could also be obtained with [DEOA][Glycolic](61.9%), [DEOA][Glycolate](59.5%) and [DEOA][Butyrate](53.3%) than [DEOA][Lactate] (48.2%), but the ratios of α/β (selectivity) in them were lower than that in [DEOA][Lactate] (Table R3). These results showed that the promoting factors for reaction selectivity might be discordant with the factors that could enhance the yields (reactivity) in the current reaction, which made it more difficult to design effective reaction catalytic systems. On the other hand, the combination of cations and anions can lead to a synergic effect for a given reaction system, which has also been shown important for the current reaction. As shown in the data of Figure R2 to R5, both of the desirable selectivity and reactivity came from not a single anion or cation, but their different combinations. Therefore, ionic liquids that contain rich of structural diversity as well as potential interactions such as electrostatic interactions, H-bond interactions and so on, will offer a useful compound pool for screening

towards the development of simple and convenient aqueous synthesis α -ADPr-N₃ directly for β -NAD⁺.

Table R1 Effects of inorganic salts and lactate salts on the reaction (supplemented experiments)

Entry ^a	IL name	α Yield	β Yield	Ratio α/β
1	-- ^b	8.5%	4.0%	2.1
2	NaCl	55.3%	6.5%	8.5
3	KCl	53.7%	6.8%	7.9
4	CaCl ₂	22.7%	2.5%	9.1
5	Sodium Lactate	58.5%	5.7%	10.2
6	Potassium Lactate	59.2%	5.1%	11.5
7	Calcium Lactate	23.6%	2.8%	8.5

^a 5:1 mole fraction water : IL. ^b only water.

Table R2 Effects of lactate-based ionic liquids on the reaction

Entry ^a	IL name	α Yield	β Yield	Ratio α/β
1	[DBU][Lactate]	35.2%	4.5%	7.8
2	[Choline][Lactate]	36.4%	3.4%	10.7
3	[TEOA][Lactate]	23.9%	1.6%	14.9
4	[DEOA][Lactate]	48.2%	2.0%	24.7
5	[DMEOA][Lactate]	30.6%	1.3%	23.5
6	[TMG][Lactate]	11.4%	1.0%	11.7

^a 5:1 mole fraction water : IL.

Table R3 Effects of DEOA-based ionic liquids on the reaction

Entry ^a	IL name	α Yield	β Yield	Ratio α/β
1	[DEOA][Hexanoate]	44.4%	20.9%	2.1
2	[DEOA][Butyrate]	53.3%	14.7%	3.6
3	[DEOA][Acetate]	61.9%	28.4%	2.2
4	[DEOA][Glycolate]	59.5%	5.1%	11.7
5	[DEOA][Lactate]	48.2%	2.0%	24.7

^a 5:1 mole fraction water : IL.

Table R4 Distribution of cations and anions of ionic liquids with the ratio of α/β over 10.

Entry ^a	IL name	α Yield	β Yield	Ratio α/β
1	[Choline][Lactate]	36.4%	3.4%	10.7
2	[TEOA][Lactate]	23.9%	1.6%	14.9
3	[DEOA][Glycolate]	59.5%	5.1%	11.7
4	[DEOA][Lactate]	48.2%	2.0%	24.7
5	[DMEOA][Acetate]	13.1%	1.1%	11.7
6	[TMG][Glycolate]	21.5%	1.8%	12.2
7	[TMG][Lactate]	11.4%	1.0%	11.7
8	[BMIM][BF ₄]	10.6%	1.0%	10.6

^a 5:1 mole fraction water : IL.

Table R5 Distribution of cations and anions of ionic liquids with α yield over 40%.

Entry ^a	IL name	α Yield	β Yield	Ratio α/β
1	[Choline][Benzoate]	51.1%	7.9%	6.5
2	[TEOA][Glycolate]	42.0%	5.1%	8.2
3	[DEOA][Hexanoate]	44.4%	20.9%	2.1
4	[DEOA][Butyrate]	53.3%	14.7%	3.6
5	[DEOA][Acetate]	61.9%	28.4%	2.2
6	[DEOA][Glycolate]	59.5%	5.1%	11.7
7	[DEOA][Lactate]	48.2%	2.0%	24.7
8	[HMIM][BF ₄]	42.4%	20.0%	2.1

^a 5:1 mole fraction water : IL.

Question-2: The choice of 5:1 mole fraction water : IL is selected based on one set of screening results, and not the IL that was finally recommended. How reasonable is this selection and why weren't other ILs of interest tested in this framework? Or, rephrased, how likely are the results from these ratios to be comparable across different IL systems.

Response: We thank the reviewer for this comment. IL([TEOA][Glycolic]) was the first ionic liquid that we found to promote the reaction to give α -ADPr-N₃ with 1.5 ratio of α/β . And then we found the aqueous solution of the ionic liquid was also beneficial to enhance the ratio of α/β . Based on the two points, we investigated the effects of content of water in ionic liquids (Table S1 in the supporting information). The choice of 5:1 mole fraction water : IL was based on the

screening results.

For the next question, the interactions of ionic liquids and the reaction constituents were complicated including electrostatic interactions, H-bond interactions and even hydrophobic interactions (Egorova, K. S., Gordeev, E. G., Ananikov, V. P. *Chem. Rev.*, 2017, 117, 7132-7189). The promoting effects of different types of ionic liquids for transform of β -NAD⁺ to α -ADPr-N₃ might come from different activation mechanisms. Thus, we agree with Reviewer#1 that the results from the ratio obtained with one single water:IL are not be comparable across different IL systems.

Figure R1 Our multi-step improvement strategy

One of the goals in the current framework is to discover a practical and useful synthetic method for convenient preparations of ADPr-N₃ directly from β -NAD⁺. Toward this goal, we run a multi-step improvement strategy including four steps and a series of discovery (Figure R1 above). We chose 5:1 mole fraction water : IL([TEOA][Glycolate]) as the reaction mediate by running Step-1, but the selectivity and yield of α -ADPr-N₃ are not so good yet (ratio of α/β = 8.2; yield of α -ADPr-N₃ = 42%). Therefore, we designed the following screening method (Step-2) with a wider scope of ionic liquids. After Step-3, we got a better ionic liquid [DEOA][Lactate] in the aqueous solution to give ratio of α/β up to 24.7, however the yield of α -ADPr-N₃ was still not

desirable (48.2%). And then we run the further improvements (Step-3) for the reaction conditions through changing reaction temperatures, substrate concentrations etc. (Table S4 in Page S16 in supporting information). After running Step-3, 89% yield of α -ADPr-N₃ could be obtained with the ratio of α/β up to 25.4 in water : IL[DEOA][Lactate] system (5:1 mole fraction water:IL[DEOA][Lactate]) at 90°C with 48 mM concentration of β -NAD⁺. This reaction condition could also be applied at 0.5-gram scales. In addition, the method can be used to other nucleotide substrates such as β -NADP, 8-Br- β -NAD⁺ and cADPr (Fig. 2F, 2G, 2H in the manuscript).

Briefly, via the above four steps of improvements, a convenient and simple reaction system for preparation of α -ADPr-N₃ was discovered. The merits of this method include the one-step reaction, readily accessible materials, easy operations in open air, high α -selectivity and yield, etc., which meet our requirements for the following syntheses of ADP-ribosylated peptides and their biological researches. At the same time, we agreed with Review#1 that 54 kinds of ionic liquid aqueous solution that were tested in this work could not cover all the possibility for the optimization parameters. Based on the current screening results, studies on the detailed interactions between ionic liquids, water and reactants with a wide range of parameters are undergoing in our lab. The researches in this issue will help us to further explore the reaction mechanisms about how different types of ionic liquid aqueous solutions promote the selective nucleophile substitution reaction of β -NAD⁺.

Question-3: The highlighted ionic liquids in Figure 2D that are indicated as being taken forward to screening are not necessarily those that show the best ratios? What governed this selection? Why were other reasonable candidates excluded ahead of those showing poorer ratios? How were the error-bars calculated (numbers of replicates are not clearly highlighted for example in the captions)?

Response: We thank the reviewer for this comment. We reorganized Fig. 2D including changing its types of bars in the revised manuscript to make it clear. The highlighted ionic liquids in Fig. 2D represented the ones that showed high α/β ratios (more than 10:1), and the components of the highlighted ionic liquids were marked using the arrows to direct their cations and anions indicated in Fig. 2B. Experiments shown in Fig 2D were performed in triplicates, and the error-bars in Fig. 2D were calculated based on these data. The goal of the screening is to discover a practical and

useful reaction system for the preparation of ADPr-N₃ directly from β -NAD⁺. As shown in Fig. 2D, the highlighted ionic liquids with the α/β ratios more than 10:1 contained 6 cations and 4 anions. It was suggested the combinations of cations and anions would be more important than to choose a special one. Thus, the screening-based method shown in current work should be useful in directing future method development in finding the new catalytic functions of ionic liquids.

Question-4: BF₄⁻ anion was screened as part of the set. Why was this, given that this anion is recognised to undergo hydrolysis in aqueous solutions? Further, these anions do not appear in the list in Figure 2B?

Response: We thank the reviewer for this comment. In the previous manuscript, we only listed the organic anions. There are no specific reasons for the BF₄⁻ anion in the current screening. In the revised manuscript, we added the inorganic anions including BF₄⁻ that used for the screening in Fig.2B to make it clear.

Question-5: Is there scope for a 1-pot process utilizing the ionic liquid to aid the click reaction?

Response: Thanks Review#1's suggestion. We attempted the 1-pot process toward utilizing the ionic liquid to aid the click reaction as shown in Figure R2 below. The typical experiment procedure is described as follows: The mixture of 0.5 mg of β -NAD⁺ (0.075 μ mol) and 2 mg of sodium azide (0.03 mmol) in the mixed solvents (100 μ L) of H₂O and [DEOA][Lactate] (mol/mol = 5:1) was heated at 90 °C for 0.5 h. And then, total 50 μ L of Peptide-1' (0.2 mg), CuSO₄ (0.3 mM), THPTA (1.2 mM) in water solution was added into the untreated reaction mixture of β -NAD⁺ and sodium azide, and then 2.5 equivalent of sodium ascorbate was added. The resulted reaction was kept for 6 hours at room temperature. The reaction was detected by analysis HPLC, and 51% yield of target Peptide-1 could be obtained by our built analysis method. However, this yield was lower than the two-step-reaction method shown in the Fig. 3C (96%), in which a desalination step was involved.

Figure R2 One-pot transformation of $\beta\text{-NAD}^+$ to $\alpha\text{-ADP-ribose}$ peptide

Question-6: How were the optimization parameters/ranges selected? From the SI this seems to have been an improvement process rather than optimization.

Response: We thank the reviewer for this comment. We agree with Review#1. Temperature, time, different ionic liquids etc. were used as the parameters for improving the reaction performances. “improvement” should be better than “optimization” for describing such experiments, and thus we changed the caption of Table S4 to “Improvements for the preparative reaction conditions of $\alpha\text{-ADPr-N}_3$ ” in Page S16 in the revised manuscript.

Question-7: How was the physiological stability of the ribosylated peptides measured and under what conditions? These data should be included in the SI.

Response: We thank the reviewer for this comment. It is reported that ADP-ribosylation marks installed on different amino acid residues have varied stability. In our study, we used a triazolyl-ADP ribose linkage to mimic the ADP-ribosylation of glutamate or aspartate residues, which are highly sensitive to NH_2OH treatment even at neutral pH (H. Hilz, *Z. Physiol. Chem.*, 1981, 362, 1415-1425; P. O. Hassa, et al., *Microbiol. Mol. Biol. Rev.*, 2006, 70, 789-829; J. Moss, et al., *J. Biol. Chem.*, 1983, 258, 6466-6470; Y. J. Zhang, et al., *Nat. Methods*. 2013, 10, 981-984). To determine the stability of our ribosylated peptides, the peptide-3 was incubated in PBS buffer (pH 7.5) containing 0.5 M NH_2OH at 37 °C. The mixture was analyzed by HPLC at different time points. The peptide remained intact even after 24-hour incubation. In addition, we also

demonstrated that our ribosylated peptide was inert to acidic or basic treatments (pH = 3.0, 5.0, 9.0, 11.0) (Figure S8 in the revised supporting information and Figure R3 below). These results indicated a satisfactory stability of the triazoly-ADP ribose linkage as a mimic of the native ADP-ribosylated glutamate or aspartate.

Figure R3 Analysis of peptide-3 stability against NH₂OH, acidic or basic treatments.

To make it clear to the readers, we have added more information in the related sentences describing and discussing the experiment result in our revised manuscript as seen “*In contrast to the natural ester-linked ADP-ribosylated peptides that are unstable in NH₂OH treatment at neutral pH($t_{1/2} < 1$ h),^{39, 41, 42} all the peptides bearing triazoly-ADPr linkages were stable for at least 24 hours in NH₂OH, as well as in acidic and basic conditions (Figure S8 in the supporting information).*” in Line 6 to Line 8 in the first paragraph in Page 7 highlighted in yellow.

Question-8: For the ionic liquids: the anion names should be IUPAC consistent (-oate for carboxylates). The ¹H NMR data look very good, but I would normally expect both a reference to literature data for each of the synthesized materials and, for known compounds, 2 pieces of spectral and 1 piece of physical data to confirm identity and purity. This should be the case for all characterized compounds, unless there are clearly justified reasons as to why these cannot be obtained.

Response: We thank the reviewer for this comment. Based on the suggestions, we changed all the anion names in the manuscript and Table S2 and Table S3 in the revised supporting information to keep consistent with IUPAC rules. For the known compounds of ionic liquids, the references were added (Reference 2 to 6 in Page S106 in the revised supporting information). For the unknown **47** ionic liquids, the data of ¹³C NMR and IR besides of ¹H NMR were supplemented. The ¹³C NMR data and IR data were in Page S24 to Page S37, and the copied of ¹³C NMR spectra were in Page S55 to Page S105 in the revised SI. ¹³C NMR data and IR data of the unknown **47** ionic liquids

were also shown in Table R6 below in this letter.

Table S6 ^{13}C NMR data and IR data of the unknown 47 ionic liquids

IL-1: [DBU][Hexanoate] ^{13}C NMR (100 MHz, D_2O) δ 182.8, 165.9, 54.1, 48.2, 38.0, 37.8, 32.6, 31.3, 28.5, 26.0, 25.8, 23.5, 22.0, 19.0, 13.6 ppm. IR: 3242, 2928, 2858, 1644, 1558, 1447, 1386, 1322, 1205, 1158, 1099, 1052, 884, 882 cm^{-1} .
IL-2: [DBU][Valerate] ^{13}C NMR (100 MHz, D_2O) δ 183.2, 165.8, 54.0, 48.1, 37.9, 37.5, 32.6, 28.4, 28.2, 25.9, 23.4, 22.1, 18.9, 13.3 ppm. cm^{-1} . IR: 2929, 2859, 1642, 1554, 1446, 1384, 1322, 1202, 1158, 1107, 689
IL-3: [DBU][Crotonate] ^{13}C NMR (100 MHz, D_2O) δ 174.4, 165.8, 140.8, 127.6, 54.0, 48.1, 37.9, 32.6, 28.4, 25.9, 23.4, 18.9, 17.1 ppm. IR: 2925, 2861, 2466, 1894, 1637, 1545, 1378, 1350, 1319, 1279, 1157, 974, 897, 715, 685, 635, 529 cm^{-1} .
IL-4: [DBU][Lactate] ^{13}C NMR (100 MHz, D_2O) δ 181.6, 165.8, 68.3, 54.0, 48.1, 37.9, 32.5, 28.4, 25.9, 23.3, 20.3, 18.9 ppm. IR: 2930, 2862, 1741, 1644, 1596, 1447, 1369, 1321, 1205, 1111, 1032, 842, 691, 529 cm^{-1} .
IL-5: [DBU][H_2PO_4] ^{13}C NMR (100 MHz, D_2O) δ 165.9, 54.1, 48.2, 37.9, 32.8, 28.4, 25.8, 23.3, 18.9 ppm. IR: 2925, 2856, 2326, 1644, 1604, 1348, 1252, 1203, 1063, 939, 865, 688, 516 cm^{-1} .
IL-6: [DBU][Propionate] ^{13}C NMR (100 MHz, D_2O) δ 184.0, 165.8, 54.0, 48.1, 37.9, 32.6, 30.7, 28.4, 25.8, 23.3, 18.9, 10.2 ppm. IR: 2929, 2861, 1641, 1556, 1460, 1387, 1359, 1322, 1284, 1203, 863, 689, 528 cm^{-1} .
IL-7: [DBU][DCA] ^{13}C NMR (100 MHz, D_2O) δ 169.5, 165.9, 68.7, 54.2, 48.3, 38.1, 32.8, 28.6, 26.1, 23.5, 19.1 ppm. IR: 2931, 1639, 1446, 1350, 1322, 1205, 1106, 984, 821, 764, 690 cm^{-1} .
IL-8: [DBU][Glycolate] ^{13}C NMR (100 MHz, D_2O) δ 178.9, 165.8, 61.1, 54.1, 48.2, 37.9, 32.6, 28.4, 25.9, 23.3, 18.9 ppm. IR: 2927, 1643, 1604, 1355, 1321, 1205, 1107, 1057, 985, 895, 689, 589, 505 cm^{-1} .
IL-9: [DBU][Benzoate] ^{13}C NMR (100 MHz, D_2O) δ 173.8, 165.4, 135.9, 131.2, 129.0, 128.2, 53.8, 47.8, 37.7, 32.4, 28.2, 25.7, 23.1, 18.7 ppm. IR: 2938, 2855, 1647, 1596, 1553, 1354, 1325, 1207, 1159, 1060, 820, 717, 691, 669, 528 cm^{-1} .
IL-10: [DBU][TFA] ^{13}C NMR (100 MHz, D_2O) δ 165.9, 162.3 (q), 116.5 (q), 54.1, 48.2, 37.9, 32.7, 28.4, 25.9, 23.3, 18.9 ppm. IR: 2934, 1687, 1641, 1448, 1324, 1195, 1162, 1108, 985, 821, 797, 717 cm^{-1} .
IL-11: [Choline][Glycolate] ^{13}C NMR (100 MHz, D_2O) δ 178.4, 67.4, 60.6, 55.6, 53.8 ppm. IR: 3191, 2853, 1590, 1480, 1358, 1079, 1006, 954, 900, 685 cm^{-1} .
IL-12: [Choline][Acetate] ^{13}C NMR (100 MHz, D_2O) δ 178.5, 67.3, 55.5, 53.8, 21.6 ppm. IR: 3144, 1568, 1481, 1384, 1330, 1139, 1089, 1008, 954, 915, 867, 640, 463 cm^{-1} .
IL-13: [Choline][Hexanoate] ^{13}C NMR (100 MHz, D_2O) δ 183.6, 67.4, 55.6, 53.8, 37.3, 31.0, 25.4, 21.8, 13.3 ppm. IR: 2928, 2858, 1567, 1481, 1384, 1304, 1091, 955, 867, 614, 555, 454 cm^{-1} .
IL-14: [Choline][Lactate] ^{13}C NMR (100 MHz, D_2O) δ 181.9, 68.5, 68.3, 55.6, 53.5, 20.1 ppm. IR: 3204, 1592, 1479, 1402, 1341, 1119, 1087, 1032, 954, 865, 772, 645, 531 cm^{-1} .
IL-15: [Choline][H_2PO_4] ^{13}C NMR (100 MHz, D_2O) δ 67.3, 55.5, 53.8 ppm. IR: 3037, 2852, 2338, 1596, 1480, 1239, 1077, 936, 865, 507 cm^{-1} .
IL-16: [Choline][Benzoate] ^{13}C NMR (100 MHz, D_2O) δ 175.0, 136.1, 131.3, 128.9, 128.4, 67.2, 55.5, 53.7 ppm. IR: 3024, 1597, 1558, 1479, 1359, 1137, 1089, 953, 826, 719, 669 cm^{-1} .
IL-17: [Choline][Crotonate] ^{13}C NMR (100 MHz, D_2O) δ 173.3, 144.2, 124.8, 67.4, 55.5, 53.8, 17.2 ppm. IR: 3024, 2854, 1656, 1558, 1480, 1358, 1286, 1242, 1090, 954, 862, 716, 555 cm^{-1} .
IL-18: [Choline][Butyrate] ^{13}C NMR (100 MHz, D_2O) δ 183.1, 67.4, 55.5, 53.8, 39.2, 19.2, 13.2 ppm. IR: 3174, 1563, 1481, 1387, 1305, 1090, 955, 868, 611 cm^{-1} .
IL-19: [Choline][Valerate] ^{13}C NMR (100 MHz, D_2O) δ 183.8, 67.4, 55.5, 53.8, 37.3, 28.1, 22.0, 13.2 ppm. IR: 3175, 2956, 2861, 1563, 1480, 1388, 1309, 1090, 954, 867, 646 cm^{-1} .

IL-20: [Choline][3,5,5-Trimethylhexanoate] ¹³ C NMR (150 MHz, D ₂ O) δ 183.4, 67.4, 55.6, 53.8, 50.3, 47.9, 30.0, 29.4, 27.5, 22.2 ppm. IR: 3021, 2949, 1566, 1464, 1379, 1363, 1107. 1075, 966, 954, 896, 631, 454 cm ⁻¹ .
IL-21: [C4-Choline][Br] ¹³ C NMR (150 MHz, D ₂ O) δ 65.4, 64.9, 55.4, 51.4, 24.0, 19.1, 12.9 ppm. IR: 3248, 3011, 2964, 2873, 1487, 1459, 1047, 1027, 991, 960, 936, 918, 903, 807, 735, 583, 548, 498, 456 cm ⁻¹ .
IL-22: [C8-Choline][Br] ¹³ C NMR (150 MHz, D ₂ O) δ 65.5, 65.0, 55.5, 51.6, 31.3, 28.6, 28.5, 25.8, 22.3, 22.2, 13.8 ppm. IR: 3339, 3245, 3017, 2948, 2918, 2852, 1491, 1479, 1468, 1415, 1376, 1341, 1092, 1060, 1047, 1001, 979, 964, 939, 926, 853, 753, 722, 603, 562, 510, 464 cm ⁻¹ .
IL-23: [C12-Choline][Br] ¹³ C NMR (100 MHz, D ₂ O) δ 65.2, 65.1, 55.4, 51.6, 31.9, 29.8, 29.7, 29.6, 29.5, 29.4, 29.1, 26.1, 22.6, 22.5, 13.9 ppm. IR: 3240, 3007, 2952, 2916, 2849, 1469, 1376, 1257, 1218, 1132, 1097, 1079, 1057, 1010, 992, 963, 918, 856, 840, 763, 719, 599, 568, 517, 500, 461, 435 cm ⁻¹ .
IL-24: [TEOA][Crotonate] ¹³ C NMR (100 MHz, D ₂ O) δ 175.8, 141.3, 127.2, 55.3, 55.0, 16.9 ppm. IR: 3096, 2853, 1655, 1553, 1475, 1442, 1365, 1289, 1246, 1207, 1101, 1062, 1035, 1011, 982, 913, 862, 831, 708, 578, 517 cm ⁻¹ .
IL-25: [TEOA][Hexanoate] ¹³ C NMR (100 MHz, D ₂ O) δ 183.9, 55.2, 55.0, 37.5, 31.0, 25.5, 21.8, 13.3 ppm. IR: 3267, 2930, 2871, 1730, 1562, 1400, 1248, 1174, 1097, 1067, 1036, 910, 881, 732, 649 cm ⁻¹ .
IL-26: [TEOA][Isovalerate] ¹³ C NMR (100 MHz, D ₂ O) δ 183.4, 55.5, 55.3, 47.2, 26.1, 21.9 ppm. IR: 3234, 2954, 2869, 1709, 1561, 1397, 1260, 1213, 1167, 1097, 1067, 1034, 913, 881, 709, 646, 433 cm ⁻¹ .
IL-27: [TEOA][Butyrate] ¹³ C NMR (100 MHz, D ₂ O) δ 183.5, 55.2, 55.1, 39.5, 19.3, 13.3 ppm. IR: 3227, 2960, 2872, 1721, 1561, 1399, 1307, 1256, 1206, 1066, 1034, 881. 672, 619, 521 cm ⁻¹ .
IL-28: [TEOA][Acetate] ¹³ C NMR (100 MHz, D ₂ O) δ 181.2, 56.1, 55.3, 23.3 ppm. IR: 3226, 2871, 1567, 1400, 1154, 1064, 1032, 913, 881, 648, 616 cm ⁻¹ .
IL-29: [TEOA][Lactate] ¹³ C NMR (100 MHz, D ₂ O) δ 181.6, 68.0, 55.2, 54.9, 20.0 ppm. IR: 3266, 2973, 2878, 1721, 1583, 1450, 1410, 1344. 1121. 1095, 1065, 1033, 916, 850, 647, 533 cm ⁻¹ .
IL-30: [TEOA][Glycolate] ¹³ C NMR (100 MHz, D ₂ O) δ 179.8, 61.2, 55.3, 55.1 ppm. IR: 3230, 2881, 1584, 1399, 1318, 1062, 1005, 912, 686, 579, 515 cm ⁻¹ .
IL-31: [TEOA][3,5,5-Trimethylhexanoate] ¹³ C NMR (150 MHz, D ₂ O) δ 181.6, 55.5, 55.3, 50.8, 47.5, 30.6, 29.6, 27.4, 22.4 ppm. IR: 3251, 2952, 2868, 1711, 1561, 1394, 1364, 1267, 1212, 1164, 1098, 1072, 1035, 912, 882, 672, 654, 452 cm ⁻¹ .
IL-32: [DEOA][Hexanoate] ¹³ C NMR (100 MHz, D ₂ O) δ 183.9, 56.5, 48.8, 37.5, 31.0, 25.5, 21.7, 13.3 ppm. IR: 2954, 2928, 2859, 1552, 1399, 1305, 1099, 1068, 962, 733, 655, 518 cm ⁻¹ .
IL-33: [DEOA][Butyrate] ¹³ C NMR (100 MHz, D ₂ O) δ 183.8, 56.4, 48.8, 39.5, 19.3, 13.2 ppm. IR: 2959, 2872, 1551, 1398, 1306, 1255, 1210, 1068, 955, 894, 791, 699, 819, 513 cm ⁻¹ .
IL-34: [DEOA][Acetate] ¹³ C NMR (100 MHz, D ₂ O) δ 181.0, 56.3, 48.7, 23.2 ppm. IR: 2843, 1554, 1397, 1335, 1066, 1016, 959, 923, 648, 616, 471 cm ⁻¹ .
IL-35: [DEOA][Glycolate] ¹³ C NMR (100 MHz, D ₂ O) δ 179.6, 61.1, 56.4, 48.8 ppm. IR: 3187, 2847, 1572, 1401, 1317, 1065, 948, 912, 687, 580, 512 cm ⁻¹ .
IL-36-L: [DEOA][Lactate] ¹³ C NMR (100 MHz, D ₂ O) δ 181.9, 68.1, 57.3, 48.9, 20.1 ppm. IR: 3226, 2971, 2871, 1735, 1573, 1449, 1408, 1122, 1069, 1035, 944, 852, 773, 656, 536 cm ⁻¹ .
IL-37: [DMEOA][Acetate] ¹³ C NMR (100 MHz, D ₂ O) δ 178.9, 58.6, 55.0, 42.6, 21.8 ppm. IR: 3268, 3028, 1572, 1474, 1397, 1258, 1080, 1013, 956, 919, 858, 799, 649, 613, 452 cm ⁻¹ .
IL-38: [DMEOA][Lactate] ¹³ C NMR (100 MHz, D ₂ O) δ 181.4, 67.9, 58.7, 55.1, 42.6, 19.9 ppm. IR: 3275, 2973, 2872, 2349, 1721, 1587, 1458, 1343, 1232, 1121, 1081, 1034, 992, 922, 849, 822, 773, 648, 436 cm ⁻¹ .
IL-39: [MEOA][Butyrate] ¹³ C NMR (100 MHz, D ₂ O) δ 183.5, 56.3, 50.3, 39.4, 32.4, 19.2, 13.2 ppm. IR: 2960, 2872, 2439, 1553, 1458, 1394, 1337, 1305, 1254, 1210, 1150, 1085, 1040, 1003, 945, 893, 791, 700, 620, 532

cm ⁻¹ .
IL-40: [MEOA][Acetate] ¹³ C NMR (100 MHz, D ₂ O) δ 180.8, 56.2, 50.3, 32.4, 23.1 ppm. IR: 2731, 2441, 1556, 1392, 1334, 1149, 1082, 1039, 1012, 918, 648, 616, 533, 470 cm ⁻¹ .
IL-41: [TMG][Isovalerate] ¹³ C NMR (100 MHz, D ₂ O) δ 182.7, 161.3, 46.9, 38.8, 26.1, 21.9 ppm. IR: 2951, 1547, 1453, 1386, 1322, 1257, 1168, 1098, 1064, 1035, 883, 726, 639, 543, 477 cm ⁻¹ .
IL-42: [TMG][TFA] ¹³ C NMR (100 MHz, D ₂ O) δ 162.5 (q), 161.3, 116.5 (q), 38.8 ppm. IR: 3097, 1669, 1600, 1567, 1410, 1197, 1169, 1180, 1063, 1037, 879, 827, 799, 717 cm ⁻¹ .
IL-43: [TMG][DCA] ¹³ C NMR (100 MHz, D ₂ O) δ 170.3, 161.3, 68.7, 39.0 ppm. IR: 3013, 1643, 1596, 1564, 1453, 1408, 1347, 1212, 1145, 1097, 1063, 1036, 909, 881 cm ⁻¹ .
IL-44: [TMG][Glycolate] ¹³ C NMR (100 MHz, D ₂ O) δ 178.4, 161.3, 60.6, 38.8 ppm. IR: 3219, 3006, 2929, 2879, 1684, 1600, 1561, 1482, 1454, 1428, 1410, 1390, 1311, 1235 cm ⁻¹ .
IL-45: [TMG][Crotonate] ¹³ C NMR (100 MHz, D ₂ O) δ 175.6, 161.3, 141.1, 127.4, 38.8, 17.0 ppm. IR: 3014, 2821, 1655, 1593, 1536, 1481, 1438, 1409, 1360, 1291, 1245, 1213, 1156, 1107 cm ⁻¹ .
IL-46: [TMG][Lactate] ¹³ C NMR (100 MHz, D ₂ O) δ 181.9, 161.3, 68.4, 38.8, 20.2 ppm. IR: 2967, 1595, 1563, 1451, 1407, 1372, 1342, 1238, 1115, 1091, 1064, 1033, 880 cm ⁻¹ .
IL-47: [TMG][Valerate] ¹³ C NMR (150 MHz, D ₂ O) δ 183.6, 161.3, 38.9, 37.1, 27.9, 22.0, 13.2 ppm. IR: 2954, 1547, 1454, 1389, 1237, 1096, 1065, 1035, 883, 727, 652, 542 cm ⁻¹ .

Question-9: In the case of the alpha- (1) and beta- (1') azides, the ¹H NMR clearly indicates impurities/isomers/residual starting material. Please indicate further what these peaks are and their impact.

Response: We thank the reviewer for this comment. We checked the ¹H NMR spectra of alpha- (1) and beta- (1') azides in the SI. The peaks at the range of 3.75 ppm to 1.0 ppm 3 ppm were assigned to -CH₂N- and CH₃CH₂- of triethylammonium that exchanged from TEAB (triethylammonium bicarbonate) buffer. The peaks at 8.0 ppm to 9.0 ppm and 5.0 ppm to 6.5 ppm in both of the ¹H NMR spectra and H-H COSY spectrum were further analyzed. The two peaks at 8.0 ppm to 9.0 ppm suggested the impurities had a purine base. The correlations of the small peaks at 6.2 ppm and 5.3 ppm with the small peaks at the range of 3.5 ppm to 4.9 ppm (Figure R4 below), suggested that the impurities including two riboses similar to ADPr unit. Thus, we deduced that the impurity might come from the reactions of starting material β-NAD⁺. Measures of their molecular weight by MALDI were unsuccessful, however, these impurities could be conveniently removed by the HPLC purification, and the clean ¹H NMR spectra were updated in the revised supporting information. The clean ¹H NMR spectra were also shown in Figure R5 below.

Figure R4 Analysis of peaks of impurities by H-H COSY spectrum

Figure R5 Copies of ^1H NMR spectra of purified α -ADPr- N_3 and β -ADPr- N_3

Question-10: Figures S7 and S9 it would help to mark the relevant cross-peaks on the spectra for clarity.

Response: We thank the reviewer for this comment. The H-H COSY spectra show the peaks at 5.0 ppm to 6.5 ppm might belong to the ribose rings as mentioned in the response of Question-9. Together with the peaks at 8.0 ppm to 9.0 ppm that might belong to the protons of a purine base, we deduced that the impurity might come from the reactions of starting material β -NAD⁺.

Question-11: For all figures in the SI, it would be helpful to have more detailed captions.

Response: We thank the reviewer for this comment. Based on the suggestion, we revised the captions of figures in the supporting information, and the resulted supporting information was reformulated with the revised figures and supplementary experiments.

Question-12: Figure S11 could be improved by angling the triazole in each case so that it doesn't clash with the proline.

Response: We thank the reviewer for this comment. The angles of triazoles were improved to avoid clashing with the prolines in Figure S7 in Page S21 in the revised supporting information.

Question-13: For the synthesis of ribosylated proteins the time is quoted for the procedure as 2-4 h. This should be more precisely identified; whether this is a per peptide difference, or some other reason. The MALDI data should have all peaks assigned.

Response: We thank the reviewer for this comment. The precise reaction times for the synthesis of each ribosylated peptides were added in detail in Table S7 (highlighted in yellow) in Page S18 the revised supporting information. Also as shown in Table R7 below, their reaction time were 4 hours for the peptide 1 and peptide 2. The reaction time were 2 hours for peptide 3 and peptide 4. The precise reaction times were obtained by detecting the reaction procedures by HPLC, and the reactions were completed based on disappearing of the peaks of the starting materials (the peptides). The MALDI data were revised to have all peaks assigned in Page S46 and Page S47 in the revised supporting information (highlighted in yellow). The copies of the MALDI spectra were also as shown in Figure R6 below).

Table R7 Results of click reactions for preparations of ADP-ribosylated peptides.

Compound	AA1	AA2	AA3	Peptide	R1	t _R ^a	Time	Yield ^a
Peptide-1	10	Pro	9 α	H2B(3-19)	Biotin	14.9min	4h	96%
Peptide-2	10	Pro	9 β	H2B(3-19)	Biotin	16.2min	4h	97%
Peptide-3	--	Pro	9 α	H2B(3-19)	--	9.3min	2h	98%
Peptide-4	--	Pro	9 β	H2B(3-19)	--	9.4min	2h	98%

^a Isolated yields.

Figure R6 Copies of MALDI-TOF-MS of Peptide 3 and Peptide 4.

Question-14: It would be helpful to link the immunoblots in Figure 4 to the experimental in the SI. Although this can be worked out (the captions are fine), it would improve the clarity for the reader.

Response: We thank the reviewer for this comment. In the revised manuscript, we linked the

experiments to the part of “methods” and the supplementary Figures in the supporting information as seen “See “*Photo-cross-linking and visualization of the biotinylated proteins*” in the part of “*Methods*” for more details of the photo-cross-linking-based assays. The uncropped blots and protein markers are provided in Figure S9 in the supporting information. The immunoblotting assays are provided in Page S9 in the supporting information.” in Page 8 below Fig. 4 in the revised manuscript, which was highlighted in yellow.

Question-15: Overall the manuscript would benefit from a thorough language editing. Currently some aspects are either hard to understand or not well specified and there are a number of typographical errors. This is less of a problem for the SI, where there are only a few minor issues, but these should also be looked at.

Response: We thank the reviewer for this comment. Based on the suggestions, we checked the typographical errors carefully in manuscript and SI. The specialized language editing was conducted for the revised manuscript.

Part-II: Response to reviewer#2

The manuscript by Zhu et al. describes a new synthetic strategy for the generation of an azidoADP-ribose analog (N_3 -ADPr). This analog contains an azide group at the 1' position of the glycosidic ring. The synthesis uses ionic liquids and the authors developed an efficient synthesis for generating the biologically relevant alpha isomer of N_3 -ADPr. They then coupled N_3 -ADPr to an H2B peptide (known ADP-ribosylation target of PARP1). The peptide also contain a benzophenone crosslinker as well as a biotin affinity handle. The authors sowed that the peptide could bind to a macro domain protein mH2A1.1, which is a protein known to recognize ADPribose. The authors also showed that this peptide could pulldown known ADP-ribose binding macro domain containing proteins (i.e. mH2A1.1 and PARP9) from HeLa cell lysates that were treated with UV light to induce crosslinking. Lastly, the authors showed that the ADPr-peptide could be ADP-ribosylated by PARP1.

Overall this is an interesting study; however, there are some major issues that need to be addressed prior to publication. These include:

Question-1: Overall, the manuscript really needs to be carefully edited. There are many grammar and spelling mistakes. Additionally, there are issues regarding scholarship (more on this below).

Response: We thank the reviewer for this comment. Based on the suggestions, we checked the typographical errors carefully in manuscript and supporting information. The specialized language editing was conducted for the revised manuscript.

Question-2: In the paper, the authors state that the half-life for Glu/Asp-ADPr is < 1 h. They ref. 31, but this ref. doesn't state this. Please provide the correct ref. for this statement or remove.

Response: We thank the reviewer for this comment. The correct reference is “ref 13: Moyle PM, Muir TW. Method for the Synthesis of Mono-ADP-ribose Conjugated Peptides. *J Am Chem Soc* 2010, 132, 15878-15880.” In this paper, the author described that “the ester-linked ADPR generated by PARPs is unstable at basic pH ($t_{1/2} < 1$ h pH 7.5)”. By tracing the literatures about the stabilities for Glu/Asp-ADPr, we found earlier reports in “Hilz, H. ADP-ribosylation of proteins: a multifunctional process. *Z. Physiol. Chem.* **362**, 1415-1425 (1981)”; “Adamietz, P. & Hilz, H. Poly(Adenosine Diphosphate Ribose) is covalently linked to nuclear proteins by two types of bonds. *Z. Physiol. Chem.* **357**, 527-534 (1976)”; “Bredehorst, R., Wielckens, K., Gartemann, A., Lengyel, H., Klapproth, K. & Hilz, H. Two different types of bonds linking single ADP-Ribose residues covalently to proteins. Quantification in eukaryotic cells. *Eur. J. Biochem.* **92**, 129-135 (1978)”; “Moss, J., Yost, D. A. & Stanley, S. J. Amino acid-specific ADP-ribosylation. *J. Biol. Chem.* **258**, 6466-6470 (1983).” and “Hassa, P. O., Haenn, S. S. et al. Nuclear ADP-ribosylation reactions in mammalian cells: where are we today and where are we going? *Microbiol. Mol. Biol. Rev.* **70**, 789-829 (2006)”. Based on these literatures, we revised the statements as seen “*Notably, the ADP-ribosylation on the glutamate is a barely accessible ADP-ribosylation mark, mainly due to the intrinsic instability of the ester-type glycosidic bond.*^{4, 39-43,}” in Line 3 to Line 4 in Paragraph 2 in Page 6 in the revised manuscript and “*In contrast to the natural ester-linked ADP-ribosylated peptides that are unstable in NH₂OH treatment at neutral pH($t_{1/2} < 1$ h),^{39, 41, 42} all the peptides bearing triazolyl-ADPr linkages were stable for at least 24 hours in NH₂OH, as well as in acidic and basic conditions (Figure S8 in the supporting information)*” in Line 16 to Line 19 in Paragraph 2 in Page 6 in the revised manuscript. All the contents and the correct references were highlighted in yellow in the revised manuscript.

Question-3: In the manuscript text, the IC₅₀ values stated in the text are molar but they should be micromolar according to Fig. 3. Also, the authors that the values they obtained are similar to literature values, but different macro domain proteins were tested (i.e. MacroD2 and TARG1 catalytic dead mutants in the ref: “Synthetic α - and β -Ser-ADPribosylated Peptides Reveal α -Ser-ADPr as the Native Epimer,” *Organic Letters* 40, 4140-4143 (2018). Also the position of the ADPr is different-they should comment on this.

Response: We thank the reviewer for pointing out the mistake in the unit of IC₅₀ value, which has been now corrected in the revised manuscript.

For the second question, we are a little bit confused, since the experiment described by the reviewer and the paper the reviewer referred to did not match with each other. For the sake of clarity, we provide a table below to summarize related information.

	ADP-ribosylated peptide	Protein	Method used and readout
Our study	Histone H2B peptide with ADP-ribosylation mark at Glu2	Macrodomain of mH2A1.1	Photo-cross-linking, EC ₅₀ = 2.1 μ M
Our reference cited: H. A. V. Kistemaker et al. , Synthesis and Macro-domain Binding of Mono-ADP-Ribosylated Peptides, Angew Chem Int Edit , 2016, 55, 10634-10638.	Histone H2B peptide with ADP-ribosylation mark at Glu2 (peptide 19 in the paper)	Macrodomain of catalytically dead MacroD2	ITC, K _d = 2.8 μ M
		Macrodomain of catalytically dead TARG1	ITC, no binding
The reference listed by the reviewer: J. Voorneveld et al. , Synthetic α - and β -Ser-ADPribosylated Peptides Reveal α -Ser-ADPr as the Native Epimer, Org Lett , 2018, 20, 4140-4143.	Histone H2B peptide with ADP-ribosylation mark at Ser6 (peptide 1 in the paper)	N.A.	N.A.

Question-4: One major issue is that the ADPr peptide they synthesized has the triazol-linked ADPr in a non-native position. The authors should synthesize a peptide that places the ADPr at the serine (Ser10 in the KS motif) that gets ADPr by PARP1, similar to what was done in *Organic*

Letters 40, 4140-4143 (2018).

Response: We thank the reviewer for raising this concern. However, we would like to point out that the Glu2 position of histone H2B is an endogenously ADP-ribosylated site. In fact, ADP-ribosylation is known to occur on multiple amino acid residues, including arginine, lysine, serine, asparagine, cysteine, as well as glutamate and aspartate. The ADP-ribosylation of histone H2B Glu2 position was detected as early as 40 years ago (L. O. Burzio, et al., *J Biol Chem*, 1979, 254, 3029-3037; N. Ogata, et al., *J Biol Chem*, 1980, 255, 7610-7615). Recent studies also provided evidence that supported the existence of this modification (see for example: G. J. Grundy et al., *Nat Comm*, 2016, 7, 12404). Among all the known ADP-ribosylation marks, those on glutamate and aspartate residues with ester-type glycosidic bonds are the most chemically unstable, whose half-life in neutral NH_2OH buffer can be as short as minutes (H. Hilz, *Z. Physiol. Chem.*, 1981, 362, 1415-1425; P. O. Hassa, et al., *Microbiol. Mol. Biol. Rev.*, 2006, 70, 789-829; J. Moss, D. A. Yost, S. J. Stanley, *J. Biol. Chem.*, 1983, 258, 6466-6470; Y., J. Zhang, et al., *Nat. Methods*. 2013, 10, 981-984). Such high lability makes it a challenge to investigate the biology functions of the ADP-ribosylation on glutamate or aspartate residues. In our study, we, therefore, focused on the histone H2B peptide. We deign the triazolyl-linked ADPr to mimic the fragile ester-linked ADPr (Figure R7 below) as a proof-of-concept study of our synthetic method. We demonstrated that the triazolyl-ADP ribose linkage as a good mimic of the ADP-ribosylated glutamate or aspartate residue at different levels, including: i) our mimic selectively bound to its known ‘readers’ with high affinity at single protein level; ii) our mimic could enrich endogenous ‘readers’ from cell lysate; and iii) our mimic could undergo poly ADP-ribosylation by PARP1.

Figure R7 Triazolyl-ADPr as stable mimic of the ester-ADPr

Question-5: The manuscript states: “And more importantly, the stable triazolyl linkage offered, to the best of our knowledge, the first class of chemical tools with controllable stereochemistry at the

glutamate-type glycosidic bond to study the functional differences between the α - and β -ADP-ribosylation epimers.” This is not accurate. See this ref: “Synthetic α - and β Ser-ADP-ribosylated Peptides Reveal α -Ser-ADPr as the Native Epimer,” *Organic Letters* 40, 4140-4143 (2018). The authors should discuss this paper.

Response: We thank the reviewer for this comment. As has been mentioned in the response for Point 4 above, ADP-ribosylation is known to occur on multiple amino acid residues, including arginine, lysine, serine, asparagine, cysteine, as well as glutamate and aspartate. Our study focused on the development of a triazolyl-ADP ribose linkage as a mimic of the ADP-ribosylated glutamate or aspartate residue, since such modifications has been challenging to study because of the intrinsic instability of the ester-type glycosidic bond. At the same time, we also acknowledge that multiple methods have been developed by the community to install ADP-ribosylation (or mimics) at different amino acid residues. We have now turned down our tones by changing the sentence to “*More importantly, the stable triazole linkage offers a class of useful chemical tools with controllable stereochemistry at the glutamate-type glycosidic bond, which can be used to study the functional differences between the α - and β -ADP-ribosylation epimers*” in Line 7 to Line 8 in the first paragraph in Page 8 in the revised manuscript highlighted in yellow. Also, in our revised manuscript, we have cited and discussed the paper listed by the reviewer as reference 44.

Question-6: Fig 1B: replace asparaginic with aspartate for consistency with glutamate.

Response: We thank the reviewer for this comment. We have replaced asparaginic with aspartate in Fig 1B in the revised manuscript.

Question-7: Fig 1: the authors should also include the previous synthesis of N₃-ADPr (*Molecules*, 2017) in Fig. 1.

Response: We thank the reviewer for this comment. Fig.1 demonstrates the strategies for synthesizing α -type ADP-ribosylated peptides in this work. Our previous work (*Molecules*, 2017) is to prepare β -ADPr-N₃, which doesn't belong to the scope of this work.

Question-8: Fig. 4: are there controls for crosslinking? Also, in Fig. 4f, the last lane should be “+” for peptide-3 competition if I understand the experiment correctly.

Response: We thank the reviewer for this comment. All the photo-cross-linking-based experiments showed in Fig. 4 were performed with controls. In Fig. 4A, we examined the efficiency of the photoaffinity peptide-1 in capturing mH2A1.1, a known ADP-ribosylation ‘reader’, by cross-linking different concentrations of peptide-1 with the protein. A sample without adding peptide-1 was also prepared, which served as the no probe control. The result showed that the peptide-1-induced labeling of mH2A1.1 followed a concentration-dependent manner with $EC_{50} = 2.1 \mu\text{M}$. No cross-linking signal was observed in the no probe control sample. In Fig. 4B, we examined the binding affinity of the peptide-1-induced labeling of mH2A1.1 by using ribosylated H2B peptide, peptide-3, as a competitor. The photo-cross-linking was performed in the presence of different concentrations of peptide-3. A sample without adding peptide-3 was also prepared as a no competitor control. The result showed that peptide-3 could effectively compete off the cross-linking with $IC_{50} = 4.0 \mu\text{M}$. In Figs. 4C and 4D, similar experimental design was used to show that the β -ADP-ribosylation epimer exhibited dramatically lowered ability to capture mH2A1.1 compared with the α epimer shown in Fig. 4A and 4B. The 4C and 4D panels serve as good controls of the 4A and 4B panels to demonstrate that the α epimer is critical for the recognition of ADP-ribosylation by mH2A1.1. In Fig. 4E, we tested the specificity of peptide-1-induced labeling toward mH2A1.1. Three different control samples were prepared, including no probe control, a loss-of-ADP-ribosylation-binding mH2A1.1 mutation (mH2A1.1 G224E) control, and a non-ADP-ribosylation ‘reader’ (mH2A1.2) control. The result showed that after UV irradiation, only the sample containing mH2A1.1 and peptide-1 led to robust cross-linking, indicating the labeling was indeed macrodomain-ADP-ribosylation recognition-dependent but not non-specific cross-linking. In Fig. 4F, we moved a step forward to determine if the endogenous ADP-ribosylation ‘readers’ could be enriched by the photoaffinity peptide-1. The cell lysate was photo-cross-linked with peptide-1. The labeled proteins were then enriched with streptavidin-coated agarose beads, which were subsequently eluted and subjected to immunoblotting analysis against antibodies of two known ADP-ribosylation ‘readers’, mH2A1.1 and PARP9. Samples without the photoaffinity peptide-1 or with the addition of peptide-3 as competitor were also prepared as controls. The result showed that both endogenous proteins could be selectively enriched by peptide-1, as no chemiluminescence signal was observed in the sample without peptide-1, while the cross-linking was effectively competed off by the addition of peptide-3. In summary, the above photo-cross-linking-based assays were designed and performed with appropriate controls. The results demonstrated that our triazoly-ADP ribose linkage served as a good mimic of the native ADP ribosylation, which could be used to study the interactions between ADP-ribosylation and its binding proteins.

In Fig. 4F, the last lane for peptide-3 should be “+”. A right version of Fig. 4 has been attached to the revised manuscript. We apologize for this mistake.

Question-9: Fig. 4: in the main text, the wrong panel is referenced in several sections.

Response: We apologize again for our carelessness during the preparation of the original manuscript. The mismatch between Fig. 4 and the text has been corrected during our revision.

Question-10: Fig 4f: Did the authors consider doing MS/MS proteomics? In this way, they could discover potentially new ADPr binding proteins. This would certainly add novelty to the manuscript as this has not been done before with other ADPr-peptides.

Response: We thank the reviewer for this comment. We agree with the reviewer that a comprehensive profiling of ADP-ribosylation binding proteins at the proteomic level is an important and interesting application of our ADP-ribosylation mimic. However, we think it is out the scope of our current manuscript, which is focusing on the development of a simple, effective, and scalable method to synthesize an azide-functionalized α -ADP-ribose analog directly from native β -NAD⁺ in clean ionic liquid systems, and also the following generation of ribosylated peptides with ‘click chemistry’. The identification of previously unknown ADP-ribosylation binding partners is one of our future directions to pursue. At the same time, we also hope the publication of our method can offer the community an easily accessible way to obtain chemical tools that match their own research interests in the study of protein ADP-ribosylation.

Question-11: Fig. 4: Should include molecular weights next to blots.

Response: We thank the reviewer for this comment. We have added the uncropped membranes together with protein markers for all the blots in Fig 4. as Figure S9 in the revised SI (also see Figure R8 below).

Figure R8 Uncropped membranes of blots listed in Fig. 4.

Question-12: Fig. 4g: see point 4. The correct peptide should be used before drawing any conclusions about the ADPr serving as a “priming” site. Additionally, the authors should use native NAD^+ and probe using commercially available poly-ADPr antibodies to show that polyADP-ribosylation occurs. And the authors need to determine if it is happening on the triazol-linked ADPr position and not another amino acid in the peptide.

Response: We thank the reviewer for this comment. As we have explained in the response for Point 4, the Glu2 position of histone H2B is a known endogenously ADP-ribosylated site.

To determine whether and to what extent could our ADP-ribosylation mimic initiate the poly ADP-ribosylation, we performed the assay in an alternative as has been suggested by the Reviewer#3. Specifically, we incubated the peptide-1 carrying both a mono-ADP-ribosylation mark and a biotin tag with native NAD^+ and PARP1. The reaction mixture was taken at different time points and subjected to streptavidin blotting analysis. The result has been provided as Fig. S10 in the revised supporting information (also see Figure R9 below), and it showed that the biotin signal expanded from the original position (~ 3 kDa) to the regions with higher molecular weight gradually, suggesting the generation of a mixture of heterogeneously poly-ADP-ribosylated species as time went by. After 1 hour of reaction, the biotin signal can be detected at a position with molecular weight higher than 20 kDa, indicating the growth of a poly-ADP-ribose chain containing at least 35 ADP units by a rough estimation (molecular weight of an ADP moiety is about 0.5 kDa).

In Fig. 4G, we examined if our ADP-ribosylation mimic could induce poly-ADP-ribosylation by

incubating peptide-3 (no biotin tag) with biotinylated NAD^+ and PARP1. At the same time, samples with peptide-3 only, without adding PARP1, or using unribosylated histone H2B peptide were also prepared as controls. The result showed that no biotin signal was detected from each of the three control samples, suggesting that the installation of extra ADP units to the peptide-3 was indeed relied on the presence of the mono-ADP-ribosylation mark but not on other amino acid residues, and such processes was catalyzed by PARP1 instead of non-enzymatic chemical reactions.

Figure R9 Histone H2B peptide carrying the ADP-ribosylation mimic could serve as the substrate of poly-ADP-ribosylation catalyzed by PARP1.

Question-13: This statement needs references: “especially for those on glutamate residues, can be removed by multiple enzymes,”...

Response: We thank the reviewer for this comment. During our revision, we found the sentence did not fit well with the context and thus removed it.

Question-14: Supplemental data: The ^1H NMR for the alpha and beta N3-ADPr contain some of the other isomer.

Response: We thank the reviewer for this comment. Based on the suggestions, we checked the ^1H NMR spectra of alpha- (1) and beta- (1') azides. The peaks at the range of 3.75 ppm to 1.0 ppm 3 ppm were assigned to $-\text{CH}_2\text{N}-$ and CH_3CH_2- of triethylammonium that exchanged from TEAB

(triethylammonium bicarbonate) buffer. The peaks at 8.0 ppm to 9.0 ppm and 5.0 ppm to 6.5 ppm in both of the ^1H NMR spectra and H-H COSY spectrum were further analyzed based on the suggestion in Question-10 of Review#1. The two peaks at 8.0 ppm to 9.0 ppm suggested the impurities had a purine base. The correlations of the small peaks at 6.2 ppm and 5.3 ppm with the small peaks at the range of 3.5 ppm to 4.9 ppm suggested that the impurities including two riboses similar to ADPr unit (Figure R3 below). Thus, we deduced that the impurity might come from the reactions of starting material $\beta\text{-NAD}^+$. Measures of their molecular weight by MALDI were unsuccessful, however, these impurities could be conveniently removed by an additional HPLC purification, and the clean ^1H NMR spectra were updated in the revised supporting information, and also shown in below Figure R4.

Figure R4 Analysis of peaks of impurities by H-H COSY spectrum

editing is conducted for the revised manuscript.

Comment #2: How were the stability of the ADP-ribosylation triazole-analogs of the peptides measured? I may have simply missed it, but this data should be added to the Supplementary Information.

Response: We thank the reviewer for this comment. It is reported that ADP-ribosylation marks installed on different amino acid residues have varied stability. In our study, we used a triazolyl-ADP ribose linkage to mimic the ADP-ribosylation of glutamate or aspartate residues, which are highly sensitive to NH_2OH treatment even at neutral pH (H. Hilz, Hoppe Seylers *Z Physiol Chem*, 1981, 362, 1415-1425; P. O. Hassa et al., *Microbiol Mol Biol Rev*, 2006, 70, 789-829; *J. Biol. Chem.*, 1983, 258:6466-6470; Zhang, Y. J. Zhang et al., *Nat. Methods*. 2013, 10, 981-984). To determine the stability of our ribosylated peptides, the peptide-3 was incubated in PBS buffer (pH 7.5) containing 0.5 M NH_2OH at 37 °C. The mixture was analyzed by HPLC at different time points. The peptide remained intact even after 24-hour incubation. In addition, we also demonstrated that our ribosylated peptide was inert to acidic or basic treatments (pH = 3.0, 5.0, 9.0, 11.0) (Figure S8 in the revised supporting information and below Figure R3). These results indicated a satisfactory stability of the triazolyl-ADP ribose linkage as a mimic of the native ADP-ribosylated glutamate or aspartate.

To make it clear to the readers, we have added more information in the related sentences describing and discussing the experiment result in our revised manuscript as seen “*In contrast to the natural ester-linked ADP-ribosylated peptides that are unstable in NH_2OH treatment at neutral pH ($t_{1/2} < 1$ h),^{39, 41, 42} all the peptides bearing triazolyl-ADPr linkages were stable for at least 24 hours in NH_2OH , as well as in acidic and basic conditions (Figure S8 in the supporting information).*” in Line 6 to Line 8 in the first paragraph in Page 7 highlighted in yellow.

Figure R3 Analysis of peptide-3 stability against NH_2OH , acidic or basic treatments.

Comment #3: The ordering and “call-outs” for most of the panels in Fig. 4 seem to be confused at places in the text and the figure legend.

Response: We apologize for the mismatch of panels in Fig. 4 and our text in the original manuscript and the confusion it has caused. The errors have been corrected in the revised version.

Comment #4: In Figure 4F, the “truth-table” indicates that peptide 3 is missing from all of the conditions. This should be fixed.

Response: We thank the reviewer for this comment. In Fig. 4F, the last lane for peptide-3 should be “+”. A right version of Fig. 4 has been attached to the revised manuscript.

Comment #6: The authors should test the poly-ribosylation of their analog further. They show in current Figure 4G that PARP1 can modify peptide 3 with biotinylated NAD. However, they do not appear to show how extensive this polymerization is. They should do this by labeling peptide 1 with regular NAD⁺ and showing the full molecular weight spread of the product by blotting for the biotin on peptide 1. This will give an idea of how long the poly-ADP ribosylation can become.

Response: We thank the reviewer for this comment. Following the reviewer’s suggestion, we incubated the peptide-1 carrying both a mono-ADP-ribosylation mark and a biotin tag with native NAD⁺ and PARP1. The reaction mixture was taken at different time points and subjected to streptavidin blotting analysis. The result has been provided as Figure S10 in the revised supporting information (also see Figure R9 below), and it showed that the biotin signal expanded from the original position (~ 3 kDa) to the regions with higher molecular weight gradually, suggesting the generation of a mixture of heterogeneously poly-ADP-ribosylated species as time went by. After 1 hour of reaction, the biotin signal can be detected at a position with molecular weight higher than 20 kDa, indicating the growth of a poly-ADP-ribose chain containing at least 35 ADP units by a rough estimation (molecular weight of an ADP moiety is about 0.5 kDa).

Figure R9 Histone H2B peptide carrying the ADP-ribosylation mimic could serve as the substrate of poly-ADP-ribosylation catalyzed by PARP1.

Comment #7: As a last characterization experiment, the authors should consider using peptide 1 to perform an unbiased proteomics experiment to demonstrate the utility of the peptides as a discovery-based technology.

Response:

We thank the reviewer for this comment. We agree with the reviewer that a comprehensive profiling of ADP-ribosylation binding proteins at the proteomic level is an important and interesting application of our ADP-ribosylation mimic. In our study, to test if our ribosylated photoaffinity probe could capture endogenous ADP-ribosylation binding proteins, lysate derived from HeLa S3 cells was cross-linked with peptide-1. After streptavidin enrichment, the eluted proteins were subjected to immunoblotting detection against the antibodies of two known ADP-ribosylation ‘readers’, mH2A1.1 and PARP9. The result showed that both endogenous proteins could be robustly and selectively enriched by peptide-1 from complex proteome, demonstrating the potential of our method in the identification of previously unknown ADP-ribosylation binding proteins. However, we think the performance of a proteomics study is out the scope of our current manuscript, which is focusing on the development of a simple, effective, and scalable method to synthesize an azide-functionalized α -ADP-ribose analog directly from native β -NAD⁺ in clean ionic liquid systems, and also the following generation of ribosylated peptides with ‘click chemistry’. The identification of previously unknown ADP-ribosylation binding partners is one of our future directions to pursue. At the same time, we also hope the publication of our method can offer the community an easily accessible way to obtain chemical tools that match their own research interests in the study of protein ADP-ribosylation.

Reviewers' comments:

Reviewer #1 (Remarks to the Author):

I thank the authors for their careful addressing of the points I raised in my previous review. It is clear that the time has been taken to carry out the relevant characterisations, and control experiments (which are interesting in themselves, so I look forward to the further reports of work in this area), and the language and details of the document have been substantially improved to a much more publishable standard.

There are still a few minor outstanding points:

1. For Question 10 of the original review, I did not make my point clearly. For what is now figures S11 and S13, it would be useful to highlight the key cross peaks that indicate one or other conformation, as per the description in the text (I think S10 should be S11 - and please cross check then the description for S14/(S13?) for consistency):

"To further identify the configuration of α/β -ADPr-N3, we run the H-H COSY and 1D- TOCSY and 1D-NOESY spectra below (Figure S11 to Figure S14). Figure S12 is 1D- TOCSY and 1D-NOESY spectra of α -ADPr-N3. As shown in Figure S10, the signal of the coupling of H-1 and H-3 on the distal ribose was detected beside of the signal of the coupling of H-1 and H-2, which suggested the same side of H-3 and H-1 in this molecule. While in the Figure S14 which is the 1D-TOCSY and 1D-NOESY spectra of β -ADPr-N3, only the signal of the coupling of H-1 and H-2 could be detected, suggesting the different side of H-3 and H-1. The configuration of the β -ADPr-N3 could also be confirmed by comparison its ¹H NMR spectrum with reported ¹H NMR spectrum of β -ADPr-N3.[10]"

2. Although it is pleasing to see the additional data for the 47 new ionic liquids, the standard data on purity//physical data is still not present - usually this would constitute microanalysis, melting points, and/or DSC/TGA. (and please, consistent chemdraw structures, especially not those with the negative charge embedded in the bond).

In each case, both for newly synthesised and existing ionic liquids, the appropriate multinuclear data should be provided. i.e. in addition to ¹H, also ¹³C and where appropriate ³¹P, ¹⁹F, ¹¹B.

3. In figure 2B the anion B-8 appears to be missing? There are also a couple of triazolyl (vs triazolyl) remaining in the text. In Table S2 ILs #37, 39, 40, 41, 44, and 45 still need to be converted to the appropriate IUPAC names.

4. Also, the now titled discussion is, as it was in the previous draft, a conclusion. I presume this has been changed to meet the general format of Nature, but this doesn't sit right as it stands. More effort needs to be made to add or move (from the results) discussive elements to this section, before ending with the concluding statements.

The authors most of my comments and concerns; however, there are a few things that still need to be addressed. These include:

1. Page 9, section "*Poly ADP-ribosylation catalyzed by PARP1*": the evidence that peptide-3 is poly-ADP-ribosylated is indirect. The authors should use one of the several commercially available poly-ADP-ribose antibodies to show that peptide-3 is indeed poly-ADP-ribosylated. Additionally, while I agree that the data support the notion that mono-ADP-ribosylation is required for PARP1-dependent ADP-ribosylation of their H2B peptide, the authors don't provide direct evidence that the ADP-ribose triazole in peptide 3 is acting as a "priming" site (i.e. that putative poly-ADP-ribose is added directly to the non-natural ADP-ribose triazole. Another possibility is that the mono-ADP-ribose at Glu2 (catalyzed by PARP3 according to Grundy et al., *Nature Communications*, 2016) of the H2B peptide induces PARP1-mediated ADP-ribosylation at a *different site* in the H2B peptide (e.g. serine 7, a known PARP1/2-mediated ADP-ribosylation site (Larsen, S. et al., *Cell Reports* **24**, 2493). The authors should explore this possibility experimentally or at least discuss this alternative hypothesis in their paper.
2. Page 6, line 169-717: "barely accessible" should be replaced. In fact a sentence that starts off like this: "The intrinsic chemical instability of Glu-ADP-ribosylation makes it difficult to study the effects of ADP-ribosylation at this site; therefore,..." might be better.
3. Page 9, line 245: remove s from ADPr.
4. Fig. 3D: Instead of "in nature" just say "natural linkage." And for the triazole-ADPr, I'd say "biomimetic linkage."
5. The authors should either use "triazolyl-ADPr" or triazole-ADPr," but not both.

Reviewer #3 (Remarks to the Author):

The authors have proactively addressed most of my concerns, and I am satisfied that proteomics is outside of the scope of this study due to the significant amount of synthetic chemistry. I do still have some questions concerning the polymerization of their ADP-ribosylation analog into poly-ADP-ribose (Figure S10). It appears to me that the majority of the polymerization is quite limited, as the major band that appears is only slightly higher in molecular weight. Do the authors have an estimation for how many units are added there? The authors should clearly indicate this in Figure S10. The extent of further elaboration seems low in comparison. The authors should clearly indicate in the text of the manuscript the what they think the major band is and that the further elaboration is low.

Matt Pratt

Point-by-Point Responses to the Reviewers' comments

We greatly appreciate the comments of the reviewers on our manuscript. Based on these comments, we have performed supplementary experiments. Thermal analysis and microanalysis of water contents of all the 47 new ionic liquids were performed. The supplementary physical and purity data from the experiments were added into the revised supporting information (Table R1 and Table R2 below, also see Table S8 and S9 in the revised Supporting Information). The experiments to detect poly ADP-ribosylations using commercial anti-poly(ADP-ribose) polymer antibody were conducted. The result showed that the region with robust poly ADP-ribosylation signal gradually expanded throughout the whole lane with increasing intensity as the reaction time became longer (Figure R3 below, also see Figure S10 in the revised Supporting Information), suggesting the generation and growth of poly ADP-ribose chain on peptide-1. We have also reorganized the corresponding context based on the suggestions of reviewers and checked the typographical errors carefully in manuscript and supporting information. All the revised sentences were marked with yellow in the revised manuscript. The details are as follows.

Part-I: Response to reviewer #1

Reviewer #1 (Remarks to the Author):

I thank the authors for their careful addressing of the points I raised in my previous review. It is clear that the time has been taken to carry out the relevant characterizations, and control experiments (which are interesting in themselves, so I look forward to the further reports of work in this area), and the language and details of the document have been substantially improved to a much more publishable standard.

There are still a few minor outstanding points:

Question-1: For Question 10 of the original review, I did not make my point clearly. For what is now figures S11 and S13, it would be useful to highlight the key cross peaks that indicate one or other conformation, as per the description in the text (I think S10 should be S11 - and please cross check then the description for S14/(S13?) for consistency): "To further identify the configuration of α/β -ADPr-N3, we run the H-H COSY and 1D- TOCSY and 1D-NOESY spectra below (Figure S11 to Figure S14). Figure S12 is 1D- TOCSY and 1D-NOESY spectra of α -ADPr-N3. As shown in Figure S10, the signal of the coupling of H-1 and H-3 on the distal ribose was detected beside of the signal of the coupling of H-1 and H-2, which suggested the same side of H-3 and H-1 in this molecule. While in the Figure S14 which is the 1D-TOCSY and 1D-NOESY spectra of β -ADPr-N3, only the signal of the coupling of H-1 and H-2 could be detected, suggesting the different side of H-3 and H-1. The configuration of the β -ADPr-N3 could also be confirmed by comparison its 1H NMR spectrum with reported 1H NMR spectrum of β -ADPr-N3.[10]"

Response: We thank the reviewer for this comment. Based on the suggestions, we highlighted the key cross peaks in the Figure S12 and Figure S14 that indicated their conformations in the revised Supporting Information, and revised the relevant description to make them clear. The revised context and figures were

also shown below. “To further identify the configuration of α/β -ADPr- N_3 , we run the H-H COSY, 1D-TOCSY and 1D-NOESY spectra below (Figure S11 to Figure S14). Figure S12 is 1D-TOCSY and 1D-NOESY spectra of α -ADPr- N_3 . Irradiation at the resonance frequency of H-1 produced significant NOEY correlations with H-2 (the peak was marked with green asterisk in Figure 12(D)) and H-3 (the peak was marked with red asterisk in Figure 12(D)). The 1D-NOESY data of α -ADPr- N_3 established the cis relationship between H-1 and H-3 on the distal ribose, which suggested the same side of H-3 and H-1 in this molecule. While in the Figure S14 which is the 1D-TOCSY and 1D-NOESY spectra of β -ADPr- N_3 , only the relationship between H-1 and H-2 could be detected (the peak was marked with purple asterisk in Figure 14(C)), suggesting the different side of H-3 and H-1 in this molecule. The configuration of the β -ADPr- N_3 could also be confirmed by comparison its ^1H NMR spectrum with reported ^1H NMR spectrum of β -ADPr- N_3 .^{[10]”}

Figure R1 (also see Figure S12 in the revised Supporting Information). Hydrogen spectrum of α -ADPr- N_3 , 1D-TOCSY and 1D-NOESY spectra. Note: (A) is ^1H NMR; (B) is the 1D-TOCSY spectrum with selective excitation of the H-1, (C) is the 1D-TOCSY spectrum with selective excitation of the H-1'; (D) is 1D-NOESY spectrum with selective excitation of the H-1.

Figure R2 (also see Figure S14 in the revised Supporting Information). Hydrogen spectrum of β -ADPr- N_3 , 1D-TOCSY and 1D-NOESY spectra. Note: (A) is 1H NMR; (B) is the 1D-TOCSY spectrum with selective excitation of the H-1; (C) is 1D-NOESY spectrum with selective excitation of the H-1.

Question-2: Although it is pleasing to see the additional data for the 47 new ionic liquids, the standard data on purity//physical data is still not present - usually this would constitute microanalysis, melting points, and/or DSC/TGA. (and please, consistent chemdraw structures, especially not those with the negative charge embedded in the bond). In each case, both for newly synthesised and existing ionic liquids, the appropriate multinuclear data should be provided. i.e. in addition to 1H , also ^{13}C and where appropriate ^{31}P , ^{19}F , ^{11}B .

Response: We thank the reviewer for this comment. Firstly, we redrew the structures of ionic liquids in the copies of 1H NMR and ^{13}C NMR to make them clear and keep consistent with structures shown in other Tables. Secondly, we run the ^{31}P NMR, ^{19}F NMR and ^{11}B NMR for both of newly synthesized and existing ionic liquids containing these atoms. The multinuclear data of the samples were provided in the revised Supporting Information which were highlighted in yellow (Page S26 to Page S41). Thirdly, we supplement the physical data for the 47 new ionic liquids in the revised supporting information (Table S1 and Table S2, also see Table S8 and Table S9 in the revised Supporting Information). Thermal analysis of 47 new ionic liquids was performed on a Netzsch STA 449 F5 using the Proteus Analysis software. The TGA data were collected at 10 K/min under N_2 (Range was 20 $^{\circ}C$ to 400 $^{\circ}C$). The differential scanning calorimetry (DSC) data were obtained simultaneously with the TGA data. Melting point temperatures were reported from DSC data. Onset exothermic temperatures reported from TGA data as determined from the step tangent. All the characteristic data of the ionic liquids including the onset exothermic temperature (T_{onset}) and melting point temperature (T_{mp}) were collected as shown in Table R1 (Table S8 in the revised Supporting Information) below. The copies of DSC/TGA curves of the 47 ionic liquids were attached at the end of this letter.

Table R1 Thermogravimetric analysis results and melting points of ionic liquids

Name of IL	T _{onset} / T _{mp} (°C)	Name of IL	T _{onset} / T _{mp} (°C)
[DBU][Hexanoate]	169.7 / - ^a	[TEOA][Hexanoate]	158.0 / - ^a
[DBU][Valerate]	237.9 / - ^a	[TEOA][Isovalerate]	166.5 / - ^a
[DBU][Crotonate]	233.2 / - ^a	[TEOA][Butyrate]	152.5 / - ^a
[DBU][Lactate]	215.7 / - ^a	[TEOA][Acetate]	175.4 / - ^a
[DBU][H ₂ PO ₄]	192.8 / - ^a	[TEOA][Lactate]	230.3 / - ^a
[DBU][Propionate]	227.9 / - ^a	[TEOA][Glycolate]	227.8 / - ^a
[DBU][DCA]	208.0 / - ^a	[TEOA][3,5,5-Trimethylhexanoate]	172.2 / - ^a
[DBU][Glycolate]	209.0 / - ^a	[DEOA][Hexanoate]	169.8 / - ^a
[DBU][Benzoate]	217.6 / - ^a	[DEOA][Butyrate]	149.6 / - ^a
[DBU][TFA]	189.3 / - ^a	[DEOA][Acetate]	169.2 / - ^a
[Choline][Glycolate]	192.4 / - ^a	[DEOA][Glycolate]	172.3 / - ^a
[Choline][Acetate]	187.0 / - ^a	[DEOA][Lactate]	234.3 / - ^a
[Choline][Hexanoate]	190.5 / - ^a	[DMEOA][Acetate]	201.0 / - ^a
[Choline][Lactate]	219.8 / - ^a	[DMEOA][Lactate]	231.8 / - ^a
[Choline][H ₂ PO ₄]	166.0 / - ^a	[MEOA][Butyrate]	172.3 / - ^a
[Choline][Benzoate]	204.7 / - ^a	[MEOA][Acetate]	110.7 / - ^a
[Choline][Crotonate]	200.1 / - ^a	[TMG][Isovalerate]	138.7 / - ^a
[Choline][Butyrate]	192.8 / - ^a	[TMG][TFA]	185.3 / - ^a
[Choline][Valerate]	190.5 / - ^a	[TMG][DCA]	177.1 / - ^a
[Choline][3,5,5-Trimethylhexanoate]	186.3 / - ^a	[TMG][Glycolate]	190.5 / - ^a
[C4-Choline][Br]	244.6 / 115.7	[TMG][Crotonate]	192.6 / - ^a
[C8-Choline][Br]	237.5 / 124.9	[TMG][Lactate]	210.9 / - ^a
[C12-Choline][Br]	232.3 / 80.2	[TMG][Valerate]	200.7 / - ^a
[TEOA][Crotonate]	167.0 / - ^a		

^a liquid at room temperature.

For the evidences of sample purity, we firstly analyzed the ¹H NMR and ¹³C NMR of 47 new ionic liquids (the copies of the spectra were contained from Page S57 to Page S107 in the Supporting Information) to confirm the 1:1 ratio of cations and anions, and no residual impurities. The ionic liquids were prepared by acid-base neutralization. Water are the sole by-products. The microanalysis of water contents of all the 47 ionic liquids was then performed by using Karl Fischer electrometric titration method after they were prepared and dried by the standard methods. Water contents of all the ionic liquid were less than 0.5%, the data were shown in Table R2 below (also see in Table S9 in the revised Supporting Information).

Table R2 Microanalysis of water contents of ionic liquids

Name of IL	Water content (ppm)	Name of IL	Water content (ppm)
[DBU][Hexanoate]	2654	[TEOA][Hexanoate]	2877
[DBU][Valerate]	2906	[TEOA][Isovalerate]	3356
[DBU][Crotonate]	3982	[TEOA][Butyrate]	3177
[DBU][Lactate]	2430	[TEOA][Acetate]	3558
[DBU][H ₂ PO ₄]	1033	[TEOA][Lactate]	3979
[DBU][Propionate]	2917	[TEOA][Glycolate]	3559
[DBU][DCA]	1853	[TEOA][3,5,5-Trimethylhexanoate]	3288
[DBU][Glycolate]	2521	[DEOA][Hexanoate]	3011
[DBU][Benzoate]	1556	[DEOA][Butyrate]	2926
[DBU][TFA]	585	[DEOA][Acetate]	3352
[Choline][Glycolate]	4113	[DEOA][Glycolate]	3955
[Choline][Acetate]	3269	[DEOA][Lactate]	3960
[Choline][Hexanoate]	4881	[DMEOA][Acetate]	3775
[Choline][Lactate]	2777	[DMEOA][Lactate]	3879
[Choline][H ₂ PO ₄]	3904	[MEOA][Butyrate]	2405
[Choline][Benzoate]	3075	[MEOA][Acetate]	3249
[Choline][Crotonate]	2669	[TMG][Isovalerate]	3588
[Choline][Butyrate]	955	[TMG][TFA]	836
[Choline][Valerate]	1278	[TMG][DCA]	1778
[Choline][3,5,5-Trimethylhexanoate]	3488	[TMG][Glycolate]	1093
[C4-Choline][Br]	1552	[TMG][Crotonate]	943
[C8-Choline][Br]	978	[TMG][Lactate]	1273
[C12-Choline][Br]	821	[TMG][Valerate]	1454
[TEOA][Crotonate]	2997		

Question-3: In figure 2B the anion B-8 appears to be missing? There are also a couple of triazolyl (vs triazolyl) remaining in the text. In Table S2 ILs #37, 39, 40, 41, 44, and 45 still need to be converted to the appropriate IUPAC names.

Response: We thank the reviewer for this comment. The structure of anion B-8 were added into the revised Figure 2B. The “triazoly” was corrected to “triazolyl” in the text. The names of anions were changed in the appropriate IUPAC names in the revised Table S2.

Question-4: Also, the now titled discussion is, as it was in the previous draft, a conclusion. I presume this has been changed to meet the general format of Nature, but this doesn't sit right as it stands. More effort needs to be made to add or move (from the results) discussive elements to this section, before ending with the concluding statements.

Response: We thank the reviewer for this comment. Based the suggestions, we re-organize the part of discussion carefully. The revised context includes a brief summary of the content of this work, and a discussive content about the significance of our synthetic chemistry in promoting biological researches of protein ADP-ribosylation. They were also shown below.

“In this work, we developed a new approach to construct α -type ADP-ribosylated peptides. Using a biomimetic ADP-ribosylation reaction with β -NAD⁺ as the ADPr donor, key intermediates (α -ADPr-N3) can be prepared in extremely simple [DEOA][Lactic] IL/water systems with 25:1 α/β selectivity and yields of around 86%. This reaction in tandem with click chemistry then offers an unprecedented modular synthesis of α -type ADP-ribosylated peptides. The concise synthetic route provides a fast access to biologically important ADP-ribosylated peptides with controllable sequences and functional modifications. Using the human histone H2B peptide as a model, we demonstrated the prepared α -type ADP-ribosylated H2B peptides not only have high affinities with intact ADP-ribosylation binding domains at the molecular level, but also have desirable cellular stability in enriching endogenous ADP-ribosylation protein partners in cell lysates. Furthermore, the peptides could be used as effective substrates to prepare more complicated site-specific poly ADP-ribosylated peptides.”

“ADP-ribosylation is a highly dynamic and reversible post-translational modification. The increasing importance of ADP-ribosylation in physiological and pathological functions demands well-defined α -type ADP-ribosylated peptides as the chemical tools for studying molecular mechanisms of various ADP-ribosylations, however accessing the peptides is challenging due to their inherent complicated structures and the lack of effective synthetic tools. The syntheses of α -type ADP-ribosylated peptides described in this work therefore supply a practical approach, by which all the reactions can be performed open air in EP tubes without the need for specialized instruments or training. Together with the other advantages including easily accessible materials, precise stereochemical control and concise synthetic routes, high yields and multidimensional bio-applications of products, the presented methods may become a powerful synthesis platform to produce general molecular tools for the study of protein ADP-ribosylation.”

Part-I: Response to reviewer #2

Reviewer #2 (Remarks to the Author):

The authors most of my comments and concerns; however, there are a few things that still need to be addressed. These include:

Question-1: Page 9, section “Poly ADP-ribosylation catalyzed by PARP1”: the evidence that peptide-3 is poly-ADP-ribosylated is indirect. The authors should use one of the several commercially available poly-ADP-ribose antibodies to show that peptide-3 is indeed poly-ADP-ribosylated. Additionally, while I agree that the data support the notion that mono-ADP-ribosylation is required for PARP1-dependent ADP-ribosylation of their H2B peptide, the authors don't provide direct evidence that the ADP-ribose triazole in peptide 3 is acting as a “priming” site (i.e. that putative poly-ADP-ribose is added directly to the non-natural ADP-ribose triazole. Another possibility is that the mono-ADP-ribose at Glu2 (catalyzed by PARP3 according to Grundy et al., Nature Communications, 2016) of the H2B peptide induces PARP1-mediated ADP-ribosylation at a different site in the H2B peptide (e.g. serine 7, a known PARP1/2-mediated ADP-ribosylation site (Larsen, S. et al., Cell Reports 24, 2493). The authors should explore this possibility experimentally or at least discuss this alternative hypothesis in their paper.

Response: We thank the reviewer for this comment. As suggested by the reviewer, we purchased one anti-poly(ADP-ribose) polymer antibody that has been used to detect poly ADP-ribosylation in many related studies from Abcam (ab14459). We performed the poly ADP-ribosylation reaction by incubating the peptide-1 with native NAD⁺ and PARP1. The reaction mixture was taken at different time points and subjected to immunoblotting analysis against the anti-poly(ADP-ribose) polymer antibody. As has been shown in Figure R3 below, intense luminescence signals were observed in high molecular weight region of the blot even at 0 min. We speculated that the strong signals came from the auto-poly ADP-ribosylation of PARP1 (see for example: I. Kameshita et al., J Biol Chem, 1984, 259, 4770-4776; M. Altmeyer et al., Nucleic Acids Res, 2009, 37, 3723-3738; Z. Tao et al., J Am Chem Soc, 2009, 131, 14258-14260; J. D. Chapman et al., J Proteome Res, 2013, 12, 1868-1880). To test this hypothesis, we splitted the 10 min reaction mixture into two samples. To one sample, SDS loading buffer was directly added. For another one, we tried to remove PARP-1 from the mixture by enriching the biotin-tagged peptide-1 with streptavidin-coated magnetic beads (Dynabeads, Invitrogen, 11206D). The enriched peptides were eluted by boiling the beads in SDS loading buffer. Immunoblotting of the two samples against anti-poly(ADP-ribose) polymer antibody (Figure R3 below) revealed that the intense luminescence at high molecular weight region could no longer be observed after Dynabeads enrichment. At the same time, signal, although weaker, spreading the whole lane stood out, suggesting the existence of different poly ADP-ribosylated species. Encouraged by this result, we repeated the poly ADP-ribosylation reaction. Samples taken at each time point were subjected to Dynabeads enrichment before immunoblotting analysis. The result (Figure R3 below, also see Figure S10 in the revised Supporting Information, replacing the previous Figure S10) showed that, as the reaction time became longer, the region with robust poly ADP-ribosylation signal gradually expanded throughout the whole lane with increasing intensity, suggesting the generation and growth of poly ADP-ribose chain on peptide-1. We hope

this new result obtained by anti-poly(ADP-ribose) polymer antibody could persuade the reviewer that the peptide carrying ADPR with our biomimetic linkage could serve as the substrate for poly ADP-ribosylation. For the site of poly ADP-ribosylation on the peptide, we admit that our data could not rule out that the Ser7 has also been modified besides the Glu2. We thank the reviewer for this thoughtful suggestion and have discussed this possibility in our revised manuscript.

Figure R3 Poly ADP-ribosylation of peptide-1 by PARP1. (A) ADP-ribosylation reaction samples taken at indicated time points were directly subjected to immunoblotting analysis against anti-poly(ADP-ribose) polymer antibody. (B) Mixture of 10 min poly ADP-ribosylation reaction was splitted into two samples, which were without or with Dynabeads enrichment, respectively, before immunoblotting analysis. (C) ADP-ribosylation reaction samples taken at indicated time points were subjected to Dynabeads enrichment prior

to immunoblotting analysis against anti-poly(ADP-ribose) polymer antibody. Sample with only Dynabeads was prepared as control.

Question-2: Page 6, line 169-171: “barely accessible” should be replaced. In fact a sentence that starts off like this: “The intrinsic chemical instability of Glu-ADP-ribosylation makes it difficult to study the effects of ADP-ribosylation at this site; therefore,…” might be better.

Response: We thank the reviewer for this comment. We have revised this sentence in the manuscript. “The intrinsic chemical instability of Glu-ADP-ribosylation makes it difficult to study the effects of ADP-ribosylation at this site;^{4, 39-43} *therefore replacing the highly labile ester-ADPr linkage in the native H2B peptides with a triazolyl-ADPr linkages might produce a stable chemical mimic peptide that could facilitate the investigation of its precise biological interactions (Fig. 3D)*” highlighted in yellow in page 6 in the revised manuscript.

Question-3: Page 9, line 245: remove s from ADPr.

Response: We thank the reviewer for this comment. We have removed “s” from ADPr, which was highlighted in yellow in the second paragraph in page 9 in the revised manuscript.

Question-4: Fig. 3D: Instead of “in nature” just say “natural linkage.” And for the triazole-ADPr, I’d say “biomimetic linkage.”

Response: We thank the reviewer for this comment. We revised Fig. 3D based on the suggestions in the revised manuscript.

Question-5: The authors should either use “triazolyl-ADPr” or triazole-ADPr,” but not both.

Response: We thank the reviewer for this comment. We revise them to “triazolyl-ADPr” in the revised manuscript.

Part-I: Response to reviewer #3

Reviewer #3 (Remarks to the Author):

The authors have proactively addressed most of my concerns, and I am satisfied that proteomics is outside of the scope of this study due to the significant amount of synthetic chemistry. I do still have some questions concerning the polymerization of their ADP-ribosylation analog into poly-ADP-ribose (Figure S10). It appears to me that the majority of the polymerization is quite limited, as the major band that appears is only slightly higher in molecular weight. Do the authors have an estimation for how many units are added there? The authors should clearly indicate this in Figure S10. The extent of further elaboration seems low in comparison. The authors should clearly indicate in the text of the manuscript the what they think the major band is and that the further elaboration is low.

Response: We thank the reviewer for this comment. We fully understand the reviewer's concern on the extending of the poly ADP-ribose chain by PARP1. To specifically detect the poly ADP-ribosylation, we purchased one anti-poly(ADP-ribose) polymer antibody (Abcam, ab14459). We performed the poly ADP-ribosylation reaction by incubating the peptide-1 with native NAD⁺ and PARP1. The reaction mixture was taken at different time points and subjected to immunoblotting analysis against the anti-poly(ADP-ribose) polymer antibody. As has been shown in Figure R3 below, intense luminescence signals were observed in high molecular weight region of the blot even at 0 min. We speculated that the strong signals came from the auto-poly ADP-ribosylation of PARP1 (see for example: I. Kameshita et al., *J Biol Chem*, 1984, 259, 4770-4776; M. Altmeyer et al., *Nucleic Acids Res*, 2009, 37, 3723-3738; Z. Tao et al., *J Am Chem Soc*, 2009, 131, 14258-14260; J. D. Chapman et al., *J Proteome Res*, 2013, 12, 1868-1880). To test this hypothesis, we splitted the 10 min reaction mixture into two samples. To one sample, SDS loading buffer was directly added. For another one, we tried to remove PARP-1 from the mixture by enriching the biotin-tagged peptide-1 with streptavidin-coated magnetic beads (Dynabeads, Invitrogen, 11206D). The enriched peptides were eluted by boiling the beads in SDS loading buffer. Immunoblotting of the two samples against anti-poly(ADP-ribose) polymer antibody (Figure R3 below) revealed that the intense luminescence at high molecular weight region could no longer be observed after Dynabeads enrichment. At the same time, signal, although weaker, spreading the whole lane stood out, suggesting the existence of different poly ADP-ribosylated species. Encouraged by this result, we repeated the poly ADP-ribosylation reaction. Samples taken at each time point were subjected to Dynabeads enrichment before immunoblotting analysis. The result (Figure R3 below, also see Figure S10 in the revised Supporting Information, replacing the previous Figure S10) showed that, as the reaction time became longer, the region with robust poly ADP-ribosylation signal gradually expanded throughout the whole lane with increasing intensity, suggesting the generation and growth of poly ADP-ribose chain on peptide-1. The presence of poly ADP-ribose signal at high molecular region clearly indicated that even hundreds of ADP-ribose units could be added to the peptide by PARP1.

In our previous result detected by streptavidin-biotin blot, it was true that the major band had only slight mass shift compared with the original peptide-1. One possible reason could be that the short H2B peptide was a less favored substrate for PARP1 compared with an intact protein, and the PARP1 enzyme showed a high efficiency in auto-modification (Figure R3 A and B). When PARP1 was removed from the mixture after the reaction, and the readout was changed to poly ADP-ribose signal itself instead of biotin signal, we could indeed observe the elaboration of the poly ADP-ribosylation reaction. To make the information simple and clear, we replaced the previous streptavidin-biotin blot by anti-poly(ADP-ribose) blot in our revised manuscript.

Figure R3. Poly ADP-ribosylation of peptide-1 by PARP1. (A) ADP-ribosylation reaction samples taken at indicated time points were directly subjected to immunoblotting analysis against anti-poly(ADP-ribose) polymer antibody. (B) Mixture of 10 min poly ADP-ribosylation reaction was splitted into two samples, which were without or with Dynabeads enrichment, respectively, before immunoblotting analysis. (C) ADP-ribosylation reaction samples taken at indicated time points were subjected to Dynabeads enrichment prior to immunoblotting analysis against anti-poly(ADP-ribose) polymer antibody. Sample with only Dynabeads was prepared as control.

The copies of DSC/TGA curves of the 47 ionic liquid

IL-1: [DBU][Hexanoate]

IL-2: [DBU][Valerate]

IL-3: [DBU][Crotonate]

IL-4: [DBU][Lactate]

IL-5: [DBU][H₂PO₄]

IL-6: [DBU][Propionate]

IL-7: [DBU][DCA]

IL-8: [DBU][Glycolate]

IL-9: [DBU][Benzoate]:

IL-10: [DBU][TFA]:

IL-11: [Choline][Glycolate]:

IL-12: [Choline][Acetate]:

IL-13: [Choline][Hexanoate]:

IL-14: [Choline][Lactate]:

IL-15: [Choline][H₂PO₄]:

IL-16: [Choline][Benzoate]:

IL-17: [Choline][Crotonate]:

IL-18: [Choline][Butyrate]:

IL-19: [Choline][Valerate]:

IL-20: [Choline][3,5,5-Trimethylhexanoate]:

IL-21: [C4-Choline][Br]:

IL-22: [C8-Choline][Br]:

IL-23: [C12-Choline][Br]:

IL-24: [TEOA][Crotonate]:

IL-25: [TEOA][Hexanoate]:

IL-26: [TEOA][Isovalerate]:

IL-27: [TEOA][Butyrate]:

IL-28: [TEOA][Acetate]:

IL-29: [TEOA][Lactate]:

IL-30: [TEOA][Glycolate]:

IL-31: [TEOA][3,5,5-Trimethylhexanoate]:

IL-32: [DEOA][Hexanoate]:

IL-33: [DEOA][Butyrate]:

IL-34: [DEOA][Acetate]:

IL-35: [DEOA][Glycolate]:

IL-36: [DEOA][Lactate]:

IL-37: [DMEOA][Acetate]:

IL-38: [DMEOA][Lactate]:

IL-39: [MEOA][Butyrate]:

IL-40: [MEOA][Acetate]:

IL-41: [TMG][Isovalerate]:

IL-42: [TMG][TFA]:

IL-43: [TMG][DCA]:

IL-44: [TMG][Glycolate]:

IL-45: [TMG][Crotonate]:

IL-46: [TMG][Lactate]:

IL-47: [TMG] [Valerate]:

REVIEWER COMMENTS

Reviewer #1 (Remarks to the Author):

I thank the authors for addressing the key comments previously.

For Q1. The cross peaks have been marked on S12/14, but not, as requested on S11/13 (the H-H COSY spectra). I think this has arisen due to changing numbering throughout submissions. The authors may still consider this for clarity.

For Q2. This has been addressed as requested, with the exception that microanalysis was meant as elemental analysis (%CHN), not water-content analysis (these numbers are useful, however). This helps to identify other impurities not detectable by the spectroscopic methods used; as an alternative a halide analysis might be appropriate.

For Q4. A simpler solution is probably to rename Results to Results and Discussion and Discussion to Conclusion if this is allowed in the Nature format, as these two paragraphs still primarily conclude the paper, and the literature context is provided with the current Results.

Reviewer #2 (Remarks to the Author):

The authors have addressed most of my concerns; however, I still feel like the evidence that the ADP-ribosylated peptide (peptide-1) is poly-ADP-ribosylated is not strong. Indeed, the 10H antibody they used shows some signal, but the signal is very weak, especially considering that they used 2 mM NAD⁺ (which is way too high, they need to use micromolar amounts of NAD⁺). If the authors really want to make the point that peptide-1 is indeed poly-ADP-ribosylated, the authors should perform seq. analysis to look at the polymer, as is standard in the field. Ideally, the authors should also generate a peptide (with the modified ADP-ribose at the Glu position) in which ser7 is mutated to alanine. If this peptide is not modified, this is still interesting and suggests that modification at one site can promote modification at another site. This kind of biological insight would elevate the paper substantially.

A minor point is that spelling issues persist: for example, ADP-ribosylation.

Reviewer #3 (Remarks to the Author):

The authors have addressed my concerns with their new Figure S10. The manuscript is appropriate for publication.

Matt Pratt

Point-by-point responses to the referees' comments

Reviewer #1 (Remarks to the Author):

Q-1: The cross peaks have been marked on S12/14, but not, as requested on S11/13 (the H-H COSY spectra). I think this has arisen due to changing numbering throughout submissions. The authors may still consider this for clarity.

Answer: Thanks review#1's suggestions. We have updated Figure S11 (previous S12) and Figure S13 (previous S14) in Page S47 and Page S48 in the revised supporting information, in which the cross peaks in the H-H COSY NMR spectra were marked. The updated Figure S11 and Figure S13 are also shown below.

Figure S11

Figure S13

Q-2: This has been addressed as requested, with the exception that microanalysis was meant as elemental analysis (%CHN), not water-content analysis (these numbers are useful, however). This helps to identify other impurities not detectable by the spectroscopic methods used; as an alternative a halide analysis might be appropriate.

Answer: Thanks for the all suggestions of Reviewer#1 about the characterizations of ionic liquids. We run the elemental analysis (%CHN) and halide analysis for the appropriate ionic liquid samples and explained the reasons about elemental analysis of other ionic liquid samples as shown below.

Although elemental analysis (%CHN) is encouraged for organic small molecules for the detections of their purity, it seems not to be the general method to test the purity of variant ionic liquids. (*Green Chem.*, **2008**, *10*, 1152-1161; the statistic results about methods for testing purity of ionic liquid samples published in *J. Am. Chem. Soc* and *J. Phys. Chem. B* from 2010-2019 were also shown below (**Figure R1**). The data resource for Figure R1 was attached at the end of this letter). All of the ionic liquids prepared in our work have been prepared and purified according to the traditional methods which have been reported in literature and well established in our own lab from 2005 (*J. Solution Chem.* 2005, *34*, 585-596; *Chem. Eng. J.* 2009, *147*, 27-35; *J. Chem. Thermodyn.*, 2011, *43*, 796-799), these methodologies have also been well accepted.

Fig R1. Methods for the detections of IL/PIL purity used by other researchers. Abbreviations: IL, ionic liquid; PIL, protonic ionic liquid; EA, elemental analysis (%CHN).

The analyses of purity of ionic liquids are based on the types of their synthetic routes. We demonstrated them in Scheme R1, Scheme R2 and Scheme R3 below. For the protonic ionic liquids (PILs) that were prepared by the route in Scheme R1, their possible impurities are solvents residues, unreacted starting materials and water absorbed from air. Both of solvents residues and unreacted materials could be detected by ^1H NMR and ^{13}C NMR spectra, which showed that there are no detectable these kinds of impurities. The water contents of the samples could be detected by which they were shown less 0.5% (w/w) after drying (Table S9 in the supporting information) by using Karl Fischer Titration method. Because of no any inorganic salts involved during

the preparative procedures of this kind of ionic liquids, no further detections including halide analysis were conducted.

Scheme R1 Method-(1) for preparations of protonic ionic liquids

Scheme R2 Method-(2) for preparations of choline-based ionic liquids

Choline-based ionic liquids were synthesized based on the route shown in Scheme R2. Briefly, choline hydroxide was prepared from choline chloride as the starting material through a strong basic anion exchange resin. The aqueous solvent of choline hydroxide could be obtained and then reacted with 1.0 equivalent of carboxylic acid at room temperature for 1 day. After the solvents were removed under reduced pressure, the choline-based ionic liquids could be obtained. The ionic liquids were further dried at 60°C for 24 hours under reduced pressure. The ionic liquids were characterized by ¹H NMR, ¹³C NMR, IR and DSC/TGA. The solvents and unreacted starting materials residues were detected using ¹H NMR and ¹³C NMR, which showed that there are no detectable these kinds of impurities. The possible halides residues coming from the preparation procedures were detected using chloride-selective electrode (*Chem. Eng. J.* 2009, 147, 27-35), the results showed that the chloride content all smaller than 0.05 mol/kg. Considering the possible effects of chloride on the nucleophilic substitution reactions between NAD⁺ and azide, we also run control experiments with choline chloride (0.05 mol kg⁻¹) in di H₂O as the reaction media. The reaction results showed no detectable differences in this control reaction with the reaction in sole di H₂O as the solvent.

Scheme R3 Method-(3) for preparations of brominated ionic liquids and imidazole-type ionic liquids.

Method-(3) (Scheme R3) is used for the preparations of brominated ionic liquids ([C4-

Choline][Br], [C8-Choline][Br] and [C12-Choline][Br]), and imidazole-type ionic liquids. In the typical procedure for Method-(3), 1.1 equivalent of bromo/chloroalkane was slowly added dropwise to 1.0 equivalent of 1-methylimidazole/2-dimethylaminoethanol in ethyl acetate in an ice bath, and refluxed at 75 °C. The aqueous phase was extracted after the completion of the reaction, and ionic liquids could be obtained when the aqueous solvent was removed. The ionic liquids bearing BF₄⁻ as the anions were prepared from the corresponding imidazolium ionic liquids. Briefly, through a strong basic anion exchange resin, the aqueous solvent of hydroxide/imidazolium type ILs could be obtained and then reacted with 1.0 equivalent of tetrafluoroboric acid at room temperature for 1 day. After the solvents were removed, the target ionic liquids were obtained. The ionic liquids were further dried at 60°C for 24 hours under reduced pressure. The brominated ionic liquids that haven't reported before are characterized by ¹H NMR, ¹³C NMR, IR and DSC/TGA. Their purity was detected by ¹H NMR and elemental analysis (%CHN), which showed that there were no detectable impurities. Water contents were measured by using Karl Fischer Titration method. The reported imidazole-type ionic liquids are identified by ¹H NMR and compared with the data in literatures (Reference 6 to 10 in the supporting information).

Besides, we attempted to run elemental analysis (%CHN) of other different kinds of ionic liquid samples from our ionic liquid library (Table R1). Interestingly, variant results were obtained for different types of ionic liquids. For the first group of ionic liquids including [TMG][DCA], [C12-Choline][Br], [C8-Choline][Br], [C4-Choline][Br] and [DBU][H₂PO₄], excellent data of EA could be given with less than 0.4% errors (Entry 1 to Entry 5). For the second group of ionic liquids including [DBU][Benzoate], [TEOA][Lactate], [TMG][Crotonate], etc. (Entry 6 to Entry 13), the EA results showed the samples contained different contents of water that might originate from the contact of ionic liquids with air during the experimental manipulations of EA. However, there is the third group of ionic liquids (Entry 14 to Entry 16). The reasonable EA results of them could not be obtained even considering the absorbed water contents, in which either N% or H% have more than 0.4% errors (highlighted in yellow in Table R1). Indeed, ¹H NMR showed there were no detectable solvent residues and unreacted starting materials, and there are no potential halides involved during their preparative procedures. To explain the possible reasons underlining this phenomenon, we turn to the physical and chemical properties of ionic liquids. To our best knowledge, the combustion properties of the ionic liquids bearing different combinations of cations and anions have been investigated (*Energy Environ. Sci.*, **2013**, 6, 699), in which at least a panel of ionic liquids are shown not to combust completely with 9.1~41% char yields. Based on these results, the undesirable EA (%CHN) results shown in Entry 14 to Entry 16 may be attributed to the incomplete combustibility of ionic liquids, since the EA(%CHN) experiments are based on the analysis of combustion products of the samples. In this regard, EA(%CHN) may not be a general analysis method for ionic liquids at the current stage. The collaborative work is still required in future, towards understanding comprehensive physicochemical properties during the combustion procedures of ionic liquids.

Table R1. Elemental analysis (%CHN) of different kinds of ionic liquid samples.

Entry	ILs	Elemental Analysis				
		Formula		C%	H%	N%
1	[TMG][DCA]	C7H15Cl2N3O2	Anal. calcd	34.44	6.19	17.21
			Exp.	34.47	6.31	17.07
2	[C12-Choline][Br]	C16H36BrNO	Anal. calcd	56.79	10.72	4.14
			Exp.	56.70	10.75	4.03
3	[C8-Choline][Br]	C12H28BrNO	Anal. calcd	51.06	10.00	4.96
			Exp.	50.88	10.03	4.65
4	[C4-Choline][Br]	C8H20BrNO	Anal. calcd	42.49	8.91	6.19
			Exp.	42.17	9.15	5.98
5	[DBU][H2PO4]	C9H19N2O4P	Anal. calcd	43.20	7.65	11.20
			Exp.	43.21	7.78	11.15
6	[DBU][Benzoate]	C16H22N2O2	Anal. calcd	70.04	8.08	10.21
		C16H22N2O2·0.9H2O	Anal. calcd	66.14	8.26	9.64
			Exp.	66.12	8.56	9.72
7	[Choline][Valerate]	C10H23NO3	Anal. calcd	58.51	11.29	6.82
		C10H23NO3·2H2O	Anal. calcd	49.77	11.28	5.85
			Exp.	49.59	11.31	5.72
8	[Choline][Lactate]	C8H19NO4	Anal. calcd	49.72	9.91	7.25
		C8H19NO4·1.8H2O	Anal. calcd	42.58	10.09	6.21
			Exp.	42.89	9.94	5.82
9	[TEOA][3,5,5-Trimethylhexanoate]	C15H32NO5	Anal. calcd	58.80	10.53	4.57
		C15H32NO5·0.4H2O	Anal. calcd	57.26	10.83	4.45
			Exp.	57.14	10.72	4.34
10	[TEOA][Lactate]	C9H20NO6	Anal. calcd	45.18	8.85	5.85
		C9H20NO6·0.7H2O	Anal. calcd	42.92	8.96	5.56
			Exp.	42.93	8.85	5.70
11	[DBU][TFA]	C11H17F3N2O2	Anal. calcd	49.62	6.44	10.52
		C11H17F3N2O2·0.6H2O	Anal. calcd	47.68	6.62	10.11
			Exp.	47.36	6.75	9.89
12	[TEOA][Lactate]	C7H17NO5	Anal. calcd	43.07	8.78	7.18
		C7H17NO5·0.8H2O	Anal. calcd	40.11	8.94	6.68
			Exp.	40.45	8.82	6.52
13	[TMG][Crotonate]	C9H19N3O2	Anal. calcd	53.71	9.52	20.88
		C9H19N3O2·1.6H2O	Anal. calcd	48.50	9.08	18.85
			Exp.	48.62	9.54	18.66
14	[TMG][Lactate]	C8H19N3O3	Anal. calcd	46.81	9.33	20.47
		C8H19N3O3·1.5H2O	Anal. calcd	41.37	9.55	18.09
			Exp.	41.77	9.41	15.78
15	[DBU][Crotonate]	C13H22N2O2	Anal. calcd	65.52	9.30	11.75
		C13H22N2O2·1.5H2O	Anal. calcd	58.84	9.50	10.56
			Exp.	58.45	6.76	10.24
16	[TMG][Isovalerate]	C10H23N3O2	Anal. calcd	55.27	10.67	19.34
		C10H23N3O2·0.8 H2O	Anal. calcd	51.83	10.70	18.13
			Exp.	51.89	10.40	13.41

The detailed description about the above three kinds of synthetic routes were updated in the supporting information (Page S4 to Page S6 under the title of “Preparative methods of the ionic liquids”). The purity and suppliers of the starting materials of ionic liquids are documented in the supporting information (Page S2 under the title of “General information and materials”). The instruments of elementary analysis and chloride analysis are also added in the “General information and materials” in page S2

in the supporting information. The elemental analysis(%CHN) data of [C12-Choline][Br], [C8-Choline][Br] and [C4-Choline][Br] were added to their characterization data in the revised supporting information.

Q-3: A simpler solution is probably to rename Results to Results and Discussion and Discussion to Conclusion if this is allowed in the Nature format, as these two paragraphs still primarily conclude the paper, and the literature context is provided with the current Results.

Answer: Thanks for the suggestion of Reviewer#1. We are glad to accept his/her solution if they are allowed in the Nature format.

Reviewer #2 (Remarks to the Author):

1. The authors have addressed most of my concerns; however, I still feel like the evidence that the ADP-ribosylated peptide (peptide-1) is poly-ADP-ribosylated is not strong. Indeed, the 10H antibody they used shows some signal, but the signal is very weak, especially considering that they used 2 mM NAD⁺ (which is way to high, they need to use micromolar amounts of NAD⁺). If the authors really want to make the point that peptide-1 is indeed poly-ADP-ribosylated, the authors should perform seq. analysis to look at the polymer, as is standard in the field. Ideally, the authors should also generate a peptide (with the modified ADP-ribose at the Glu position) in which ser7 is mutated an alanine. If this peptide is not modified, this is still interesting and suggests that modification at one site can promote modification at another site. This kind of biological insight would elevate the paper substantially.

Response: We thank the reviewer for this comment. We fully understand the reviewer's concerns on: 1) if the peptide with the triazolyl-ADPr mark can really be poly-ADP-ribosylated; and 2) if the poly-ADP-ribosylation chain indeed grows on the preexisting triazolyl-ADPr site but not on other residues of the peptide, such as Ser 7.

For the first concern, we do believe the signals detected by the anti-poly(ADP-ribose) polymer antibody were indeed from the poly-ADP-ribosylation. As has been mentioned in our previous response letter, the anti-poly(ADP-ribose) polymer antibody (Abcam, ab14459) has been used to detect poly ADP-ribosylation in many related studies. Just from the Abcam website (<https://www.abcam.com/poly-adp-ribose-polymer-antibody-10h-ab14459.html>), there have already been more than 30 tracking records of publications citing this antibody for different applications, including immunoblotting. The specificity of the antibody has been repeatedly examined and validated by practices in independent studies from different research groups.

In our previous poly-ADP-ribosylation assay with peptide-1, excessive NAD⁺ (2 mM) was used to make sure enough amount of ADP-ribose donors in the reaction system for the poly-ADP-ribosylation chains to grow. During our revision, we carried out the poly-ADP-ribosylation assay with a series of different concentrations of NAD⁺ (from 0 to 2

mM). Immunoblotting analysis (Fig. R2 below, and also Fig. S10B in the revised SI) of the reaction mixtures against the anti-poly(ADP-ribose) polymer antibody showed that the poly-ADP-ribosylation signal went stronger as the NAD^+ concentration increased. Specifically, the presence of the poly-ADP-ribose chain could be detected from the samples with as low as 62.5 μM and 125 μM NAD^+ , concentrations that were less than ten-fold of the substrate peptide-1 (15 μM). This result, together with our previous result using 2 mM NAD^+ but different reaction times (Fig. S10A in the revised SI), demonstrated that PARP1 catalyzed the poly-ADP-ribosylation of peptide-1 in both a time-dependent and a NAD^+ concentration-dependent manners.

Fig R2. The poly-ADP-ribosylation of peptide-1 by PARP1 is NAD^+ concentration-dependent. Peptide-1 (15 μM) in reaction buffer (50 mM Tris pH 8.0, 10 mM MgCl_2 , 1 mM DTT) containing 1 \times activated DNA was added indicated concentrations of NAD^+ and PARP1 (1 μL , 10 units/ μL). Total volume of each reaction was 20 μL . The reactions were incubated for 60 min. After the reaction, the mixtures were subjected to Dynabeads enrichment prior to immunoblotting analysis against anti-poly(ADP-ribose) polymer antibody. Sample with Dynabeads only was prepared as control.

In fact, in another poly-ADP-ribosylation assay using 15 μM peptide-3 (with no biotin tag) and 10 μM biotin-tagged NAD^+ (Fig. 4G in the manuscript), the biotin signal could be detected after the reaction, indicating the introduction of the biotinylated ADP-ribose to the substrate peptide-3 by PARP1.

In all, the above results clearly showed that the peptides with our triazolyl-ADPr marks could serve as the substrates of PARP1-catalyzed poly-ADP-ribosylation reaction, even only limited ADP-ribose donors were provided.

For the second concern, we are aware that besides the Glu 2, other amino acid residues in the histone H2B N-terminal tail could also be poly-ADP-ribosylated by PARP

enzymes, among which the Ser 7 and Ser 15 residues are the two major reported modification sites (S. C. Larsen *et al.*, *Cell Rep*, 2018, 24, 2493-2505). To determine if the poly-ADP-ribosylation catalyzed by PARP1 occurs at the preexisting triazolyl-ADPr mark or at other Ser residues, we synthesized a new peptide-1 analog (peptide-5), in which the Ser 7 and Ser 15 were mutated to Ala residues that could not be decorated by (poly-)ADP-ribosylation. We next repeated the poly-ADP-ribosylation reaction using both peptide-1 and peptide-5 in parallel. The resulting mixtures were subjected to immunoblotting analysis using the anti-poly(ADP-ribose) polymer antibody. The result (Fig. R3 below, and also Fig. S11 in the revised SI) showed that both peptides offered comparable levels of overall chemiluminescence signals, suggesting that the two peptides were decorated by similar extents of poly-ADP-ribosylation under the same conditions. Although this observation could not entirely rule out the possibility that the poly-ADP-ribosylation also took place on the Ser 7 and/or Ser 15 residues of the H2B peptide, it did support that the preexisting triazolyl-ADPr mark was preferred by PARP1 to put on extra ADP-ribose units as the dual Ser-to-Ala mutations did not lead to significant differences.

Fig R3. The preexisting triazolyl-ADPr mark is the preferred modification site of the PARP1-catalyzed poly-ADP-ribosylation. (A) Structure of peptide-5. Highlighted in red are the original Ser 7 and Ser 15 residues mutated to Ala. (B) Comparison of

peptide-1 and peptide-5 in poly-ADP-ribosylation reactions. The peptides (15 μM) in reaction buffer (50 mM Tris pH 8.0, 10 mM MgCl_2 , 1 mM DTT) containing $1\times$ activated DNA was added indicated concentrations of NAD^+ and PARP1 (1 μL , 10 units/ μL). Total volume of each reaction was 20 μL . The reactions were incubated for 60 min. After the reaction, the mixtures were subjected to Dynabeads enrichment prior to immunoblotting analysis against anti-poly(ADP-ribose) polymer antibody. Sample with Dynabeads only was prepared as control.

We have added the new data and discussions related to the poly-ADP-ribosylation reaction to our revised manuscript and SI. We do hope that these new results could persuade the reviewer that the peptides, or more specifically, our newly developed ADPr mark with the biomimetic triazolyl-linkage, could serve as the substrates of the PARP1-catalyzed poly-ADP-ribosylation.

2. A minor point is that spelling issues persist: for example, ADP-ribosylation.

Response: We are sorry for the typos. We have corrected them in the revised manuscript.

Data resource for Fig R1. The papers were retrieved from all the published work in *J. Am. Chem. Soc.* from 2010 to 2019 with “ionic liquid” as the key word in their titles; The papers were retrieved from all the published work in *J. Phys. Chem. B* from 2010 to 2019 with protonic ionic liquid as the key word in their titles.

IL type		Characterizations and purity detections					DOI No. of the paper
PIL	APIL	NMR	Water contents	EA	Others	Commercial Resources	
		--	--	--	--		10.1021/jacs.9b07134
							10.1021/jacs.8b09513
							10.1021/jacs.8b05802
							10.1021/jacs.8b03503
							10.1021/jacs.7b07731
							10.1021/jacs.7b09156
					X-ray		10.1021/jacs.7b07611
							10.1021/jacs.7b05759
							10.1021/jacs.7b04920
							10.1021/jacs.7b03357
							10.1021/jacs.7b03036
					HPLC-MS, etc.		10.1021/jacs.6b12011
					FT-IR		10.1021/jacs.6b10695
							10.1021/jacs.6b08895
							10.1021/jacs.6b05174

					HRMS		10.1021/jacs.6b02607
							10.1021/jacs.5b13425
							10.1021/jacs.5b11031
							10.1021/jacs.5b03958
							10.1021/ja508628q
							10.1021/ja508222m
							10.1021/ja506553r
							10.1021/ja506724w
							10.1021/ja503807r
							10.1021/ja502527y
							10.1021/ja5001517
							10.1021/ja411981c
							10.1021/ja410745a
							10.1021/ja406230f
							10.1021/ja405032c
							10.1021/ja3108593
							10.1021/ja402100r
							10.1021/ja310889z
							10.1021/ja400280f
							10.1021/ja4012645
							10.1021/ja308975e
							10.1021/ja307455w
							10.1021/ja307407e
							10.1021/ja3052816
							10.1021/ja303552p
							10.1021/ja211662k
							10.1021/ja208265x
							10.1021/ja207458b
							10.1021/ja2096865
							10.1021/ja203412v
							10.1021/ja2054429
							10.1021/ja204808h
							10.1021/ja108039j
							10.1021/ja105631j
							10.1021/ja1066948
							10.1021/ja107751k
					X-ray		10.1021/ja107483g
							10.1021/ja101649b
							10.1021/ja9106397
							10.1021/ja1021816
							10.1021/ja909305t
							10.1021/ja906975z
					FT-IR		10.1021/acs.jpcc.9b08032

						10.1021/acs.jpcc.9b09278
						10.1021/acs.jpcc.9b07914
						10.1021/acs.jpcc.9b03185
						10.1021/acs.jpcc.9b01274
						10.1021/acs.jpcc.8b02981
						10.1021/acs.jpcc.7b12455
						10.1021/acs.jpcc.7b10671
						10.1021/acs.jpcc.7b10701
						10.1021/acs.jpcc.7b07945
						10.1021/acs.jpcc.7b06376
						10.1021/acs.jpcc.7b03658
				DSC and TG		10.1021/acs.jpcc.7b01309
						10.1021/acs.jpcc.6b11624
						10.1021/acs.jpcc.6b10562
						10.1021/acs.jpcc.6b09298
						10.1021/acs.jpcc.6b01203
				IC		10.1021/acs.jpcc.6b01422
						10.1021/acs.jpcc.5b02853
						10.1021/acs.jpcc.5b00274
						10.1021/jp510691e
						10.1021/jp509557z
						10.1021/jp507408r
						10.1021/jp504998t
						10.1021/jp412352k
						10.1021/jp5019179
						10.1021/jp500281z
						10.1021/jp407715e
						10.1021/jp4043232
						10.1021/jp310927w
						10.1021/jp312241h
						10.1021/jp3052714
						10.1021/jp306110g
						10.1021/jp305230r
						10.1021/jp303761f
						10.1021/jp304735p
						10.1021/jp3044237
						10.1021/jp3010844
						10.1021/jp205580w
						10.1021/jp2078727
						10.1021/jp203204c
						10.1021/jp200233q
				FT-IR		10.1021/jp1112203
						10.1021/jp105087n

REVIEWERS' COMMENTS

Reviewer #2 (Remarks to the Author):

While sequence analysis would be idea for showing polymer formation, the new Ser to Ala modified peptide data is a very nice addition. I support publication.